# The GPCR antagonist PPTN synergizes with caspofungin providing increased fungicidal activity against *Aspergillus fumigatus*

Thaila Fernanda dos Reis,[1,2] Endrews Delbaje,[1] Camila Figueiredo Pinzan,[1] Rafael Bastos,[2,3] Suzanne Ackloo,[4] Sara Fallah,[5] Bradley Laflamme,[5] Nicole Robbins,[5] Leah E. Cowen,[5] Gustavo H. Goldman[1,2]

**ABSTRACT**  Fungal pathogens pose a serious threat to human health, with *Candida* and *Aspergillus* spp. representing some of the most significant opportunistic invaders. *Aspergillus fumigatus* causes aspergillosis, one of the most prevalent fungal diseases of humans. There is a limited number of drugs available to combat these infections, and antifungal drug resistance is on the rise. In this manuscript, we show 4-[4-(4-Piperidinyl) phenyl]-7-[4-(-(trifluoromethyl) phenyl]-2-naphthalenecarboxylic acid (PPTN), a highly specific antagonist of the human P2Y14 receptor, is a promising antifungal adjuvant against diverse fungal pathogens. PPTN interacts with caspofungin (CAS), ibrexafungerp, voriconazole (VOR), and amphotericin against *A. fumigatus* CAS- and VOR-resistant clinical isolates, and also CAS against *Candida* spp and *Cryptococcus neoformans*. The combination of PPTN and CAS increases cell death in *A. fumigatus*. In the model yeast *Saccharomyces cerevisiae,* heterozygous deletion of genes involved in chromatin remodeling results in PPTN hypersensitivity, and in *A. fumigatus*, PPTN can have increased fungicidal activity when combined with the histone deacetylase inhibitor trichostatin A and the DNA methyltransferase inhibitor 5-azacytidine. Finally, PPTN has reduced toxicity to human immortalized cell lineages and partially clears *A. fumigatus* conidia infection in A549 pulmonary epithelial cells. Our results indicate that PPTN is a novel adjuvant antifungal drug against fungal diseases caused by *A. fumigatus* and *Candida* spp.

**IMPORTANCE**  Invasive fungal infections have a high mortality rate, causing more deaths annually than tuberculosis or malaria. *Aspergillus fumigatus* is the main etiological agent of aspergillosis, one of the most prevalent and deadly fungal diseases. There are few therapeutic options for treating this disease, and treatment commonly fails due to host complications or the emergence of antifungal resistance. Drug repurposing, where existing drugs are deployed for other clinical indications, has increasingly been used in the process of drug discovery. Here, we show that 4-[4-(4-Piperidinyl) phenyl]-7-[4-(-(trifluoromethyl) phenyl]-2-naphthalenecarboxylic acid (PPTN), a highly specific antagonist of the human P2Y14 receptor, when combined with caspofungin (CAS), ibrexafungerp, voriconazole (VOR), and amphotericin can increase the fungicidal activity against not only *A. fumigatus* CAS- and VOR-resistant clinical isolates but also CAS against *Candida* spp.

**KEYWORDS**  *Aspergillus fumigatus*, caspofungin, PPTN, *Candida* spp, *Cryptococcus neoformans*

The *Aspergillus* genus is composed of more than 300 species of filamentous fungi that are widely dispersed in the environment. *Aspergillus* species are saprophytes

**Peer Reviewer** W. Scott Moye-Rowley, The University of Iowa, Iowa City, Iowa, USA

Address correspondence to Gustavo H. Goldman, ggoldman@usp.br.

The authors declare no conflict of interest.

See the funding table on p. 19.

that are found in the soil and decaying plant matter where they play important roles in carbon and nitrogen recycling (1–3). Moreover, *Aspergillus spp.* are associated with animals and plants, surviving as commensal or pathogenic organisms on or within these hosts. Of clinical concern, *Aspergillus spp.* are the etiological agents of aspergillosis, a group of heterogeneous clinical conditions, that range from superficial infections to severe invasive and disseminated infections, resulting in the death of thousands of people every year (4–7). Aspergillosis triggers several clinical conditions with variable prevalence and mortality, depending on the condition of the host and their immune system (8). *Aspergillus fumigatus* is a thermotolerant opportunistic human pathogen and the most prevalent species causing invasive pulmonary aspergillosis (IPA), known as the most severe pathology among aspergillosis clinical diseases (5, 9). IPA mainly affects human hosts with severe immunosuppression and underlying lung disease, and may, depending on the host's immunological condition, reach a mortality rate of up to 90% (8, 10–13).

The treatment of aspergillosis is based on the administration of a limited armamentarium of antifungals. First-line treatment for IPA relies on the administration of broad-spectrum triazoles such as itraconazole, posaconazole, and voriconazole (VOR), with VOR being the drug of choice (14, 15). Triazoles target and inhibit the 14α‑demethylase enzyme, impairing biosynthesis of ergosterol, the principal sterol component of the fungal cell membrane. Once this pathway is blocked, there is an accumulation of 14-methylated sterols leading to alterations in membrane fluidity with consequent disruption of cell membrane integrity and inhibition of fungal growth (16–20). An increasing concern is the fact that azoles are employed not only in the clinic but are also widely used in the veterinary and agricultural fields. This has been directly linked to the widespread emergence of azole-resistant isolates among human fungal pathogens (21–26). Thus, the failure of azole therapy associated with the rapid emergence of resistant fungal strains has imposed a very challenging scenario in the treatment of invasive clinical mycoses such as IPA. As a second-line therapy, polyenes, such as liposomal amphotericin B (AMPHO), and echinocandins can be used under certain situations. Their limited use is mainly related to the high levels of toxicity (in the case of the AMPHO) and the fungistatic activity against *A. fumigatus* (in the case of echinocandins). In addition, these treatments commonly fail due to host complication issues or the emergence of antifungal resistance (14, 27–34).

The search for new antifungal drugs is critical, and drug repurposing, where approved drugs are utilized for other clinical indications, has been increasingly used to accelerate the discovery of new therapeutics (35–42). Recently, our group explored the repurposing of drugs in combination with the echinocandin caspofungin (CAS) to identify new therapeutic strategies against *A. fumigatus* (43). Through these efforts, we identified the 4-[4-(4-Piperidinyl) phenyl]-7-[4-(-(trifluoromethyl) phenyl]-2-naphthalenecarboxylic acid (PPTN), a highly specific antagonist of the human P2Y14 receptor, as a CAS synergizer (43). Here, we characterize PPTN as potentiating the antifungal effects of CAS not only against *A. fumigatus* but also against other fungal pathogens.

## RESULTS

### The GPCR antagonist PPTN when combined with CAS increased fungicidal activity against *A. fumigatus*

Recently, our group identified 12 compounds that enhance CAS activity against *A. fumigatus* (43). Out of these 12 compounds, the G protein-coupled receptor (GPCR) antagonist PPTN (Fig. 1a) reduced *A. fumigatus* metabolic activity by 56% when 20 µM of PPTN was combined with a sub-inhibitory concentration of CAS (43). PPTN antagonizes P2Y receptors, which are a family of G protein-coupled receptors (GPCRs) that are activated by extracellular nucleotides. More specifically, PPTN is a highly selective $P2Y_{14}R$ antagonist (44–46) and until recently had never been reported to have antifungal activity.

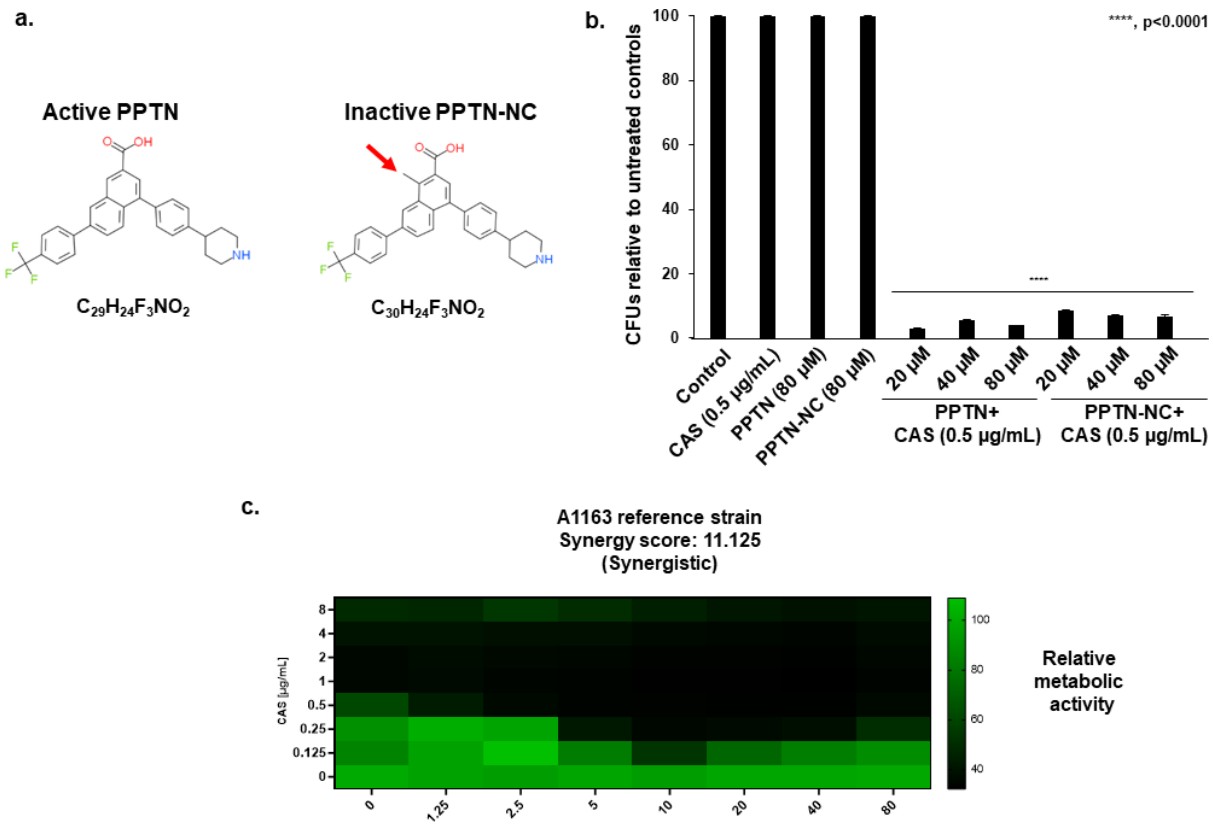

**FIG 1** PPTN and its inactivated form NC-PPTN show synergistic antifungal activity against *A. fumigatus* when combined with CAS. (a) Molecular structures of active PPTN and inactive NC-PPTN. (b) *A. fumigatus* conidia were exposed to PPTN, CAS, PPTN + CAS, NC-PPTN, NC-PPTN + CAS for 48 h at 37°C in liquid MM. Then, the cultures were plated on MM, and colony-forming units (CFUs) were assessed. The results are expressed as CFUs relative to untreated controls and are the average of three replicates ±standard deviation (****$P$ < 0.0001). (c) Heat map illustrating the checkerboard's results of PPTN (0–80 µM) combined with CAS (0–8 µg/mL) against *A. fumigatus*.

To get more insights into the antifungal activity of PPTN, we checked the minimum inhibitory concentration (MIC) against *A. fumigatus*. When *A. fumigatus* was exposed to PPTN alone, the drug was not able to inhibit fungal growth up to 80 µM (Fig. 1). Accordingly, the MIC value of PPTN was defined as higher than 80 µM. To check whether the combination of PPTN with CAS results in fungicidal or fungistatic activity against *A. fumigatus,* we inoculated *A. fumigatus* conidia in minimal medium (MM) containing PPTN alone, CAS alone, or the combination in a 96-well plate. After a 48 h incubation at 37°C, no visual growth was observed in the wells containing the compound combination. To check whether the combination treatment resulted in a fungistatic or fungicidal effect, we plated the whole volume of each well on solid MM to check for cell viability (Fig. 1b). CAS in combination with all concentrations of PPTN examined reduced *A. fumigatus* conidial viability by more than 90% (Fig. 1b), demonstrating that the activity between these two compounds is fungicidal against *A. fumigatus*.

To evaluate whether the activity of PPTN was dependent on the ability of the drug to interact with a fungal GPCR, we took advantage of an inactive counterpart of PPTN called NC-PPTN (negative control—PPTN, https://pubchem.ncbi.nlm.nih.gov/compound/ 155561056), a modified version of the compound that is 4,000 times less potent than the active form as a P2Y$_{14}$R antagonist (Fig. 1a). Similar to what was described previously, *A. fumigatus* conidia were inoculated in MM supplemented with NC-PPTN, CAS, or the combination in a 96-well plate for 48 h at 37°C. Interestingly, CAS in combination with NC-PPTN promoted fungal killing to the same extent (Fig. 1b). These results suggest that the *A. fumigatus* PPTN target is divergent from the mammalian target.

To further interrogate the chemical interaction between PPTN and CAS, we performed a checkerboard assay with CAS and PPTN (Fig. 1c). When combined with PPTN at concentrations greater than 5 µM, a lower concentration of CAS (0.25 µg/mL) substantially reduced fungal metabolic activity. Given that PPTN does not have a defined MIC, we were not able to define the fractional inhibitory concentration (FIC) value of this drug combination. However, a synergy score was calculated using the SynergyFinder system (https://synergyfinder.fimm.fi) and showed that CAS and PPTN have a synergistic effect with a synergy score of 11.125 (Fig. 1c).

To determine whether PPTN interacts with other antifungal agents, we assessed, through a checkerboard assay, the interaction between PPTN and amphotericin B (AMPHO), ibrexafungerp (IBX), and voriconazole (VOR). Using the SynergyFinder software, PPTN showed a synergistic interaction with AMPHO, IBX, and VOR against *A. fumigatus* (synergy score values larger than 10) (Fig. 2a through c). Altogether, these results suggest that although PPTN alone has no antifungal effect on its own, when combined with several classes of antifungals, it can increase the fungicidal activity against *A. fumigatus in vitro*.

Subsequently, we assessed whether the combination of CAS with PPTN would be effective against *A. fumigatus* CAS- or voriconazole (VOR)-resistant *CYP51*-dependent and -independent clinical isolates (that have VOR MICs >2.0 µg/mL; Table 1). The MIC of PPTN and the minimum effective concentration (MEC) for CAS were defined for 24

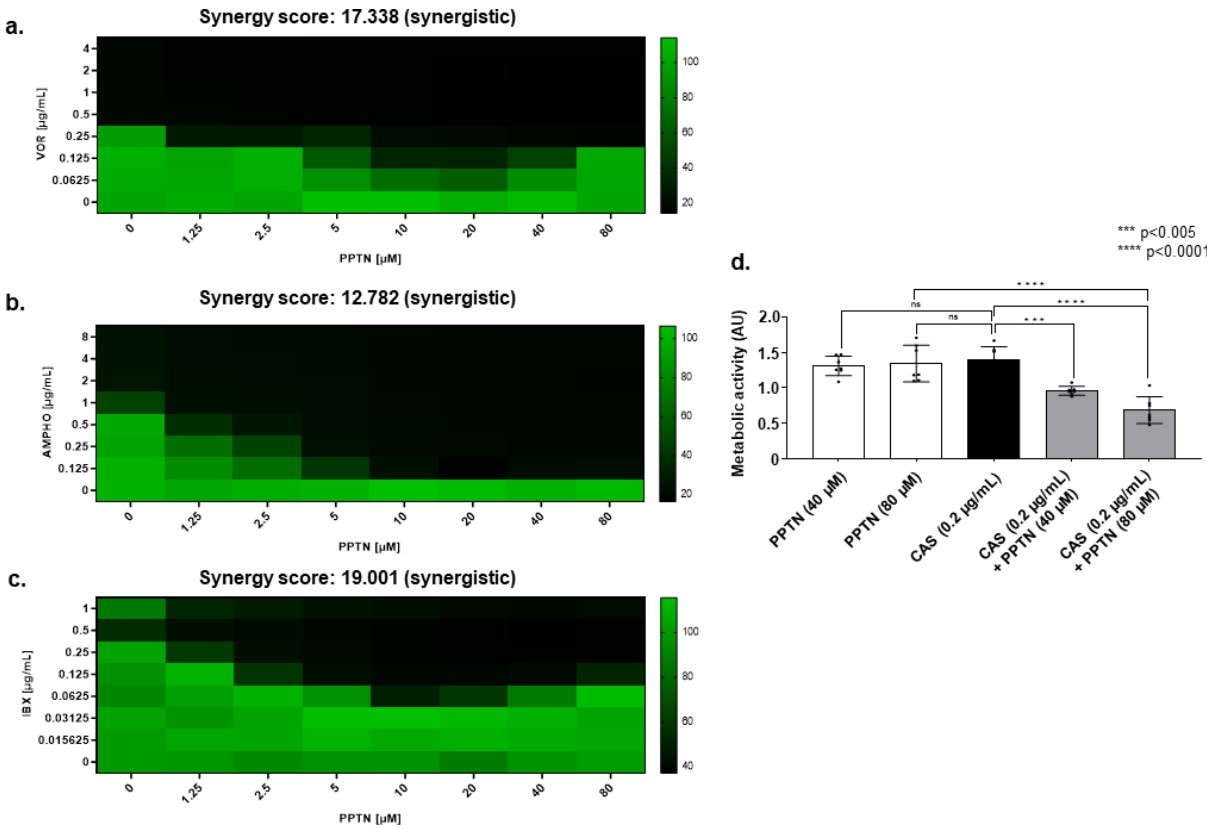

FIG 2 Metabolic activity of checkerboard assays of PPTN in combination with IBX, AMPHO, and VOR. The synergy scores for PPTN x VOR. (a) PPTN x AMPHO (b.), and PPTN x IBX (c.) were determined by the analysis of the checkerboard data using the SynergyFinder software. *A. fumigatus* was grown in liquid MM using 96-well plates at 37°C in the presence of different concentrations of the selected drugs, and after 48 h the % of metabolic activity was assessed with Alamar blue by reading fluorescence at a wavelength of 590 nm emission/530 nm excitation. A synergy score less than −10 suggests an antagonistic interaction, from −10 to 10 suggests an additive interaction, and larger than 10 suggests a synergistic interaction. The results represent the average ± standard deviation of three independent experiments. (d) Metabolic activity measurement expressed by XTT of a 24 h *A. fumigatus* biofilm and treated for an additional 12 h with CAS (0.2 µg/mL), PPTN (40 and 80 µM,) or PPTN (40 and 80 µM) combined with CAS (0.2 µg/mL). The graphic shows the value of 9 technical replicates from three independent experiments ±SD ($n = 3$; two-way ANOVA, Tukey's post-test;*$P$ <0.05, ***$P$ < 0.001, and ****$P$ < 0.0001).

*A. fumigatus* strains (23 clinical isolates resistant to VOR plus the reference strain). All isolates tested presented a MEC of CAS of 0.25 µg/mL, while all but two strains (CYP-15–213 and CYP-15–215) showed full growth up to 80 µM PPTN (Table 1). Nevertheless, when a sub-MEC of PPTN (10 µM) was combined with a sub-inhibitory concentration of CAS (0.2 µg/mL), the growth of all 24 strains was visually inhibited. We also examined compound efficacy against three *A. fumigatus* isolates presenting resistance to CAS (MEC 16 µg/mL): DPL1035, MD24053, and CM7555. The first two isolates are strains harboring mutations in the *fks1* gene, encoding the β−1,3-glucan synthase, and the last strain has an unknown mechanism of resistance. Similarly, the combination of PPTN (20 µM) with a sub-MEC of CAS (0.2 µg/mL) inhibited fungal growth (Table 1). These results suggest that the combination of PPTN with CAS is equally effective against *A. fumigatus* voriconazole-resistant isolates as it is against voriconazole-susceptible isolates.

Finally, *A. fumigatus* biofilms are notorious for their extreme tolerance to antifungal drugs (47, 48). Thus, we wanted to assess whether the synergy observed between CAS and PPTN against *A. fumigatus* was maintained when the organism was in a biofilm state. While the addition of the single agents alone resulted in no significant reduction of metabolic activity of the biofilm formation (Fig. 2d), the combination of CAS with PPTN resulted in significant reductions of the metabolic activity of a preformed biofilm relative to individual treatments (Fig. 2d). Interestingly, biofilm-phase growth requires PPTN higher drug levels for inhibition than are needed for planktonic cells. Thus, CAS and PPTN are effective at inhibiting *A. fumigatus* growth under both planktonic and biofilm conditions.

**TABLE 1** MECs and MICs for CAS and PPTN

| Strains | MEC CAS (µg/mL) | MIC PPTN (µM) | PPTN (20 µM ) + caspofungin (0.2 µg/mL) |
|---|---|---|---|
| A1163 (reference strain) | 0.25 | >80 | -[b] |
| DPL1035 (CAS-resistant) | >8 | >80 | - |
| MD (caspofungin-resistant) | >8 | >80 | - |
| CM7555 (CAS-resistant) | >8 | >80 | |
| VOR-resistant (mutations in the *CYP51* TR34/L98H)[a] | | | |
| CYP-15-184 | 0.25 | >80 | - |
| CYP-15-190 | 0.25 | >80 | - |
| CYP-15-192 | 0.25 | >80 | - |
| CYP-15-195 | 0.25 | >80 | - |
| CYP-15-202 | 0.25 | >80 | - |
| CYP-15-213 | 0.25 | 20 | - |
| CYP-15-220 | 0.25 | >80 | - |
| CYP-15-221 | 0.25 | >80 | - |
| CYP-15-222 | 0.25 | >80 | - |
| CYP-15-228 | 0.25 | >80 | - |
| CYP-15-229 | 0.25 | >80 | - |
| CYP-15-230 | 0.25 | >80 | - |
| CYP-15-231 | 0.25 | >80 | - |
| VOR-resistant (*CYP51*-independent mutations) | | | |
| CYP-15-212 | 0.25 | >80 | - |
| CYP-15-215 | 0.25 | 20 | - |
| CYP-15-224 | 0.25 | >80 | - |
| CYP-15-225 | 0.25 | >80 | - |
| CYP-15-226 | 0.25 | >80 | - |
| CYP-15-108 | 0.25 | >80 | - |
| CYP-15-109 | 0.25 | >80 | - |
| CYP-15-147 | 0.25 | >80 | - |

[a]TR34 = tandem repeat mutations in the *CYP51* promoter region and L98H, replacement in the *CYP51*-coding region.
[b]-, growth inhibition.

## PPTN induces *A. fumigatus* cell death via increased permeabilization

It is already known that the activation of fungal metacaspases, for example during oxidative stress, induces markers of apoptosis-like cell death such as nuclear condensation, disorganization of the histone complex, and DNA double-strand breaks, which coincide with the loss of fungal cell viability (49). A fluorescent histone 2A construct (h2A::mRFP) has been used as a marker of cell death in *A. fumigatus* (49). To check whether the PPTN + CAS compound combination induced apoptosis-like cell death, germlings of the *A. fumigatus* h2A::mRFP strain were exposed to $H_2O_2$ (5 mM), CAS (0.125 µg/mL), PPTN (20 µM), or CAS with PPTN for 1 h. The nuclei were stained with Hoescht. The $H_2O_2$-positive control showed a complete loss of red fluorescence, indicative of a disorganized histone complex. However, no differences in mRFP fluorescence were observed in the germlings exposed to either CAS, PPTN, or the combination (Fig. 3a). We also increased the time of drug exposure to 4 h as well as increased the concentration of PPTN (80 µM). However, no differences in red fluorescence were observed (data not shown). This suggests that PPTN and CAS do not induce apoptosis-like cell death.

The fungal mitochondria are a tubular network dispersed throughout the fungal cell and can be fragmented in the presence of antifungals and oxidative stressor agents (50, 51). To check whether the addition of PPTN and CAS would affect mitochondrial integrity, we constructed an *A. fumigatus* strain expressing a mitochondria-targeted GFP. Germlings of this strain were incubated with CAS, PPTN, or the combination for 1 h at 30°C. While the control treatment with $H_2O_2$ resulted in significant mitochondrial fragmentation, all other treatments had no discernible effect on mitochondrial integrity (Fig. 3b).

Germlings of the *A. fumigatus* wild-type strain were treated with CAS, PPTN, or the combination for 1 h. Afterward, the cells were stained with propidium iodide (PI), a cell impermeant dye that only enters cells with compromised membrane integrity. Upon exposure to CAS alone, about 35% of the germlings were PI[+], while less than 10% of the cells were PI[+] upon PPTN treatment alone (Fig. 3c). However, when the cells were exposed to CAS and PPTN, more than 80% of the cells were PI[+] (Fig. 3c), suggesting that the combination treatment leads to increased cell death in *A. fumigatus*.

## *Saccharomyces cerevisiae* haploinsufficiency profiling analysis reveals that heterozygous deletion of genes involved in chromatin remodeling confers hypersensitivity to PPTN

To elucidate the antifungal mode of action of PPTN, we utilized the model yeast *S. cerevisiae* to perform haploinsufficiency profiling (HIP), a technique based on the principle that reducing the gene dosage of a compound target (or a gene important for buffering the effects of the compound) should confer hypersensitivity to the compound (52, 53). We utilized a pooled library of 968 *S. cerevisiae* heterozygous deletion mutants developed in a drug-sensitized genetic background (*pdr1Δ/Δ pdr3Δ/Δ snq2Δ/Δ*, or *3Δ*) (54, 55). The strains in this library are individually barcoded, enabling the relative abundance of individual strains to be quantified by high-throughput sequencing. By comparing relative strain abundance in the presence of a compound versus the absence of a compound, hypersensitive heterozygous mutants are identified. For all these analyses, we considered strains as depleted when the calculated average chemical-genetic (CG) value was greater or less than 6 median absolute deviations below the median. This resulted in 17 mutants that were depleted in the presence of PPTN (Table S1; https://doi.org/10.6084/m9.figshare.28152551) (Fig. 4). The HIP signature of PPTN featured significant depletion of many heterozygous mutants for genes associated with transcription and chromatin remodeling, including *ARP7* and *ARP9* (components of both the SWI/SNF and RSC chromatin remodeling complexes), as well as *TAF5* and *TAF6* (subunits of TFIID and SAGA complexes involved in RNA polymerase II transcription initiation and chromatin modification) (*P*-value 0.00078; Fig. 4a and c; Table S2; https://doi.org/10.6084/m9.figshare.28152551). Notably, heterozygous deletion of several genes

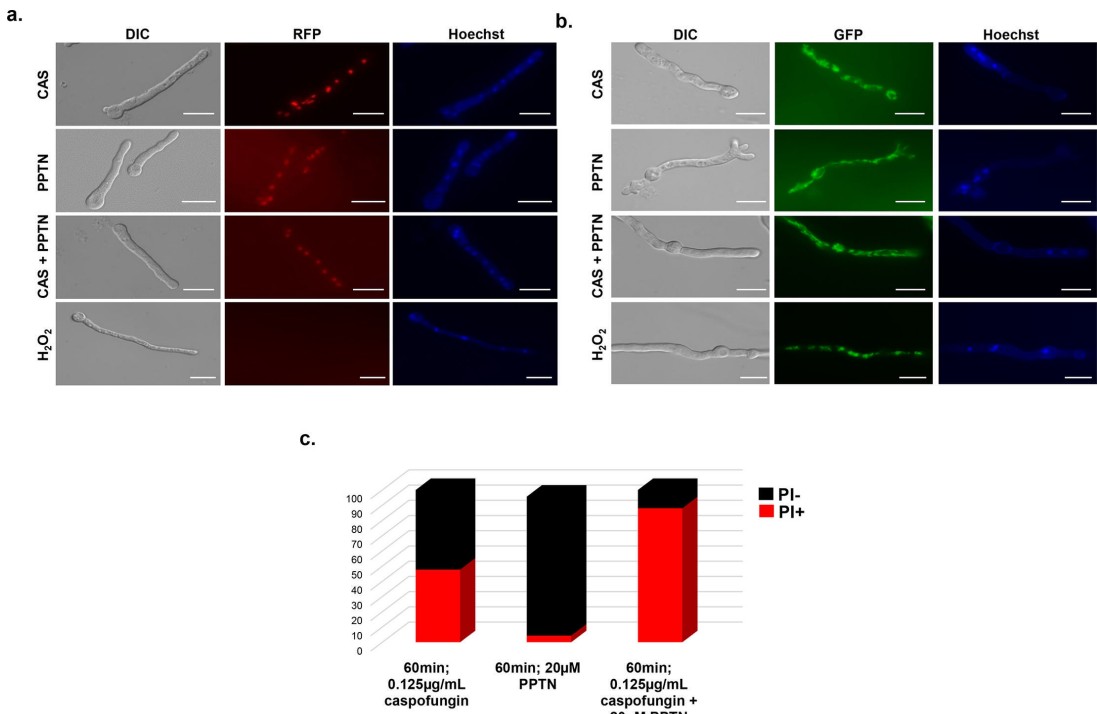

**FIG 3** PPTN +CAS induces *A. fumigatus* cell death via increased permeabilization but not apoptosis-like programmed cell death. (a) Fluorescence microscopy of *A. fumigatus* conidia containing the histone h2A::RFP exposed to CAS (0.125 µg/mL), PPTN (20 µM), CAS + PPTN (0.125 µg/mL and 20 µM, respectively) or H$_2$O$_2$ (5 mM) for 1 h at 30°C. In each experiment, 50 germlings were assessed for Hoechst staining and RFP. Scale bar = 10 µm. (b) Fluorescence microscopy of *A. fumigatus* conidia containing a mitochondrial protein tagged to GFP (mito::GFP) exposed to CAS (0.125 µg/mL), PPTN (20 µM), CAS +PPTN (0.125 µg/mL and 20 µM, respectively) or H$_2$O$_2$ (5 mM) for 1 h at 30°C. Scale bar = 10 µm. (c) Total number of stained nuclei for Hoechst and Propidium Iodide (PI)-positive in *A. fumigatus* hyphae exposed (0.125 µg/mL), PPTN (20 µM), or CAS + PPTN (0.125 + 20 µM) for 1 h. In each experiment, 50 germlings were assessed for Hoechst and PI staining.

associated with the regulation of chromatin was also identified, including heterozygous mutants for *POL2* (catalytic subunit of DNA polymerase (II) epsilon) and *ACT1* (actin, structural protein in cytoskeleton; monomeric actin in the nucleus plays a role in INO80 chromatin remodeling [56]), suggesting that genes involved in regulation of chromatin could also impact PPTN activity (Table S1; https://doi.org/10.6084/m9.fig-share.28152551). We individually reconfirmed the hypersensitivity of each heterozygous mutant strain via growth curve analysis, observing particularly strong hypersensitivity for the *ARP7, ARP9, TAF5,* and *TAF6* heterozygous mutants (Fig. 4b).

## Inhibitors of histone acetylation and DNA methyltransferase have synergistic interaction with PPTN against *A. fumigatus*

As an initial step to assess whether chromatin remodeling is involved in the mechanism of action of PPTN against *A. fumigatus*, we looked at the compound interaction between PPTN and molecules that impact gene expression. Specifically, we used three histone deacetylase inhibitors (suberoyl *bis*-hydroxamic acid [SBHA], suberoylanilide hydroxamic acid [SAHA], and trichostatin A [TSA]); one DNA methyltransferase inhibitor (5-azacy-tidine (5-AZA); and one histone acetyltransferase inhibitor (anacardic acid) (57). We observed synergistic interactions only between PPTN with trichostatin A and PPTN with 5-AZA (Fig. 5a and b; Fig. S1; https://doi.org/10.6084/m9.figshare.28152551). Intriguingly, not all histone deacetylase inhibitors showed synergy with PPTN against *A. fumigatus*, likely due to the different effects of the inhibitors on gene expression, chromatin dynamics, and/or compound affinity for its target. Overall, this suggests inhibition of

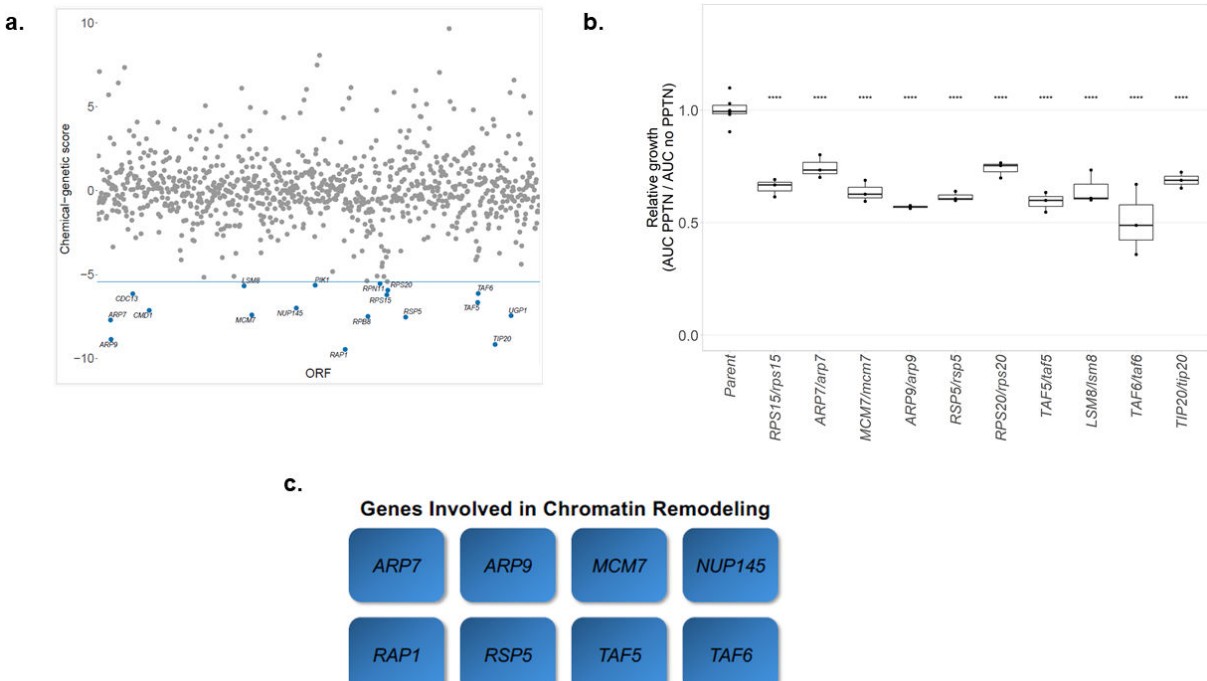

**FIG 4** Haploinsufficiency profiling in *S. cerevisiae* with PPTN. (a) Haploinsufficiency profiling with PPTN identifies many mutants implicated in chromatin remodeling as hypersensitive to PPTN treatment. A pooled barcoded library of 964 *S. cerevisiae 3Δ* heterozygous deletion strains was grown in the presence and absence of 6.25 µM PPTN for 48 hr. The relative abundance of each strain in the pool was then quantified via high-throughput sequencing of PCR-amplified barcodes, and BEAN-counter was used to calculate chemical-genetic scores for each strain in the presence of PPTN relative to a DMSO control. Significantly depleted strains (>8× median absolute deviation (MAD)) are named, with italicized gene names corresponding to the heterozygous deletion mutant. (b) Individual validation of heterozygous mutants in the presence of PPTN via growth curve analysis. Heterozygous mutants were grown both in the presence and absence of PPTN, alongside the parental *3Δ* strain. The area under the curve (AUC) was calculated for each strain in the presence and absence of PPTN, and a ratio was calculated ($AUC_{PPTN}/AUC_{No\ drug}$). All AUC ratios were normalized to the parent strain (100% relative growth) to highlight the hypersensitivity of individual mutants in the presence of PPTN. Significance was assessed based on a two-tailed unpaired *t* test adjusted for multiple comparisons using a Bonferroni multiple test correction (****$P < .0001$). Only heterozygous mutants that had a statistically significant growth deficit relative to the parental strain in at least 2 out of 3 biological replicates are highlighted. (c) Eight out of 17 genes identified in the HIP plot as important for PPTN susceptibility are involved in chromatin remodeling and are listed in rectangles.

gene expression with select histone deacetylase inhibitors and with a DNA methyltransferase inhibitor enhances the efficacy of PPTN.

## Transcriptional profiling of *A. fumigatus* exposed to PPTN and CAS

Transcriptional profiling by RNA-seq was used to identify genes for which expression was modulated by the combination of PPTN with CAS. To do so, we grew *A. fumigatus* for 16 h and exposed cultures to CAS (0.25 µg/mL), PPTN (20 µM), and CAS with PPTN at the single-agent concentrations for 1 h. Differentially expressed genes (DEGs) were defined as those with a minimum of twofold change in gene expression ($\log_2 FC \geq 1.0$ and $\leq -1.0$; FDR of 0.05) when compared to the compound-free control. By comparing those DEGs identified in the various conditions, we noted 93 and 75 genes uniquely upregulated or downregulated in the combination treatment relative to individual treatments (Fig. 6a). FunCat enrichment analyses comparing the genes differentially expressed in CAS, PPTN, and the drug combination showed overlapping categories, suggesting PPTN potentiates (positively or negatively) the gene expression changes (Fig. 6b through d). When category enrichment was considered only for the compound combination, we observed 238 upregulated genes that included processes related to ribosomal proteins, anaerobic respiration, cellular import, and non-vesicular cellular import (Fig. 6c). By contrast, there were 401 downregulated genes that encompassed

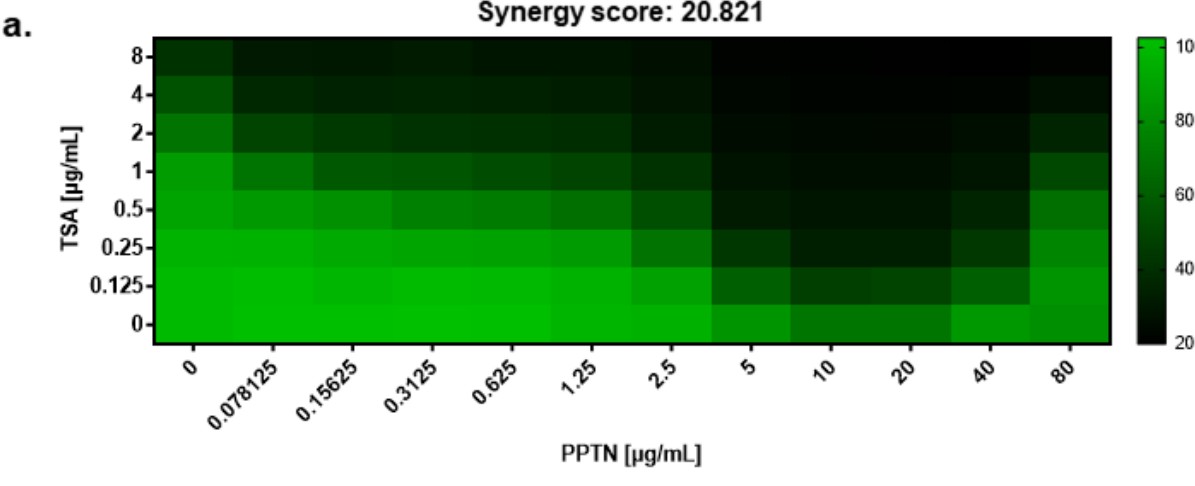

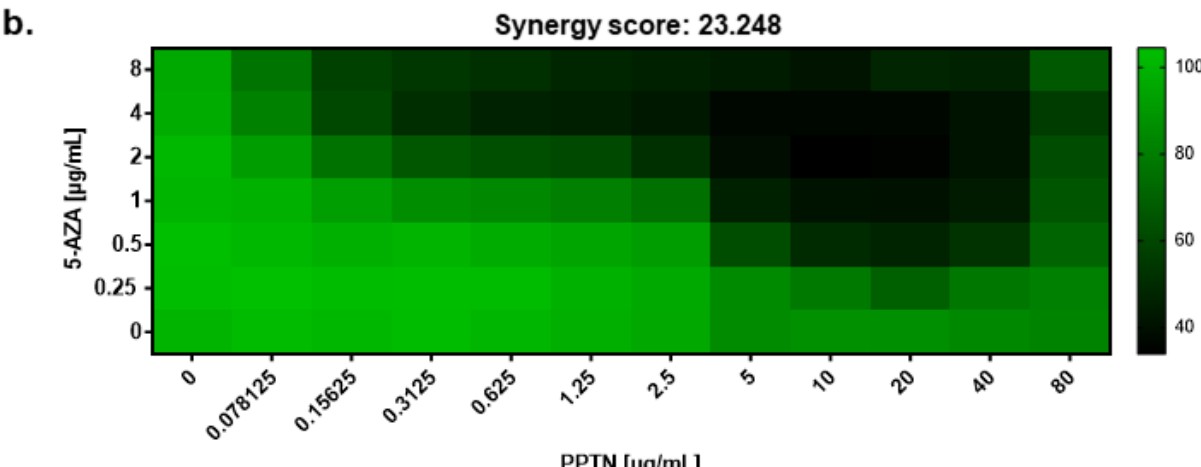

**FIG 5** Synergy between PPTN and both trichostatin A and 5-azacytidine demonstrated by checkerboard assay with a read-out of metabolic activity. The synergy scores for PPTN x Trichostatin A (a.) and PPTN x 5-azacytidine (5-AZA) (b.) were determined by the analysis of the checkerboard data using the SynergyFinder software. *A. fumigatus* was grown in liquid MM using 96-well plates at 37°C in the presence of different concentrations of the selected drugs, and after 48 h the % of metabolic activity was assessed with Alamar blue by reading fluorescence at a wavelength of 590 nm emission/530 nm excitation. A synergy score less than −10 suggests an antagonistic interaction, from −10 to 10 suggests an additive interaction, and larger than 10 suggests a synergistic interaction. The results represent the average ±standard deviation of three independent experiments.

processes related to transport ATPases, ABC transporters, secondary metabolism, lipid/ fatty acid transport, and detoxification by export (Fig. 6c). More specifically, visual inspection of the genes differentially expressed upon CAS with PPTN showed downregulation of fatty acid synthase alpha subunit *fasA* (AFUB_043770), sphinganine hydroxylase *sur2* (AFUB_016240), phospholipid metabolism enzyme regulator (AFUB_024090), cAMP-mediated signaling protein *sok1* (AFUB_064370), protein phosphatase regulatory subunit *gac1* (AFUB_029480), and H/K ATPase alpha subunit (AFUB_049200) (Fig. 6c). In addition, CAS with PPTN treatment resulted in upregulation of genes encoding a cell surface protein *mas1* (AFUB_085960), integral membrane proteins Pth-11-like (AFUB_047180 and AFUB_090700), and a C6 transcription factor (AFUB_045580). These results suggest that PPTN and CAS can modulate the expression of several genes important for transport and the integrity of the cell membrane.

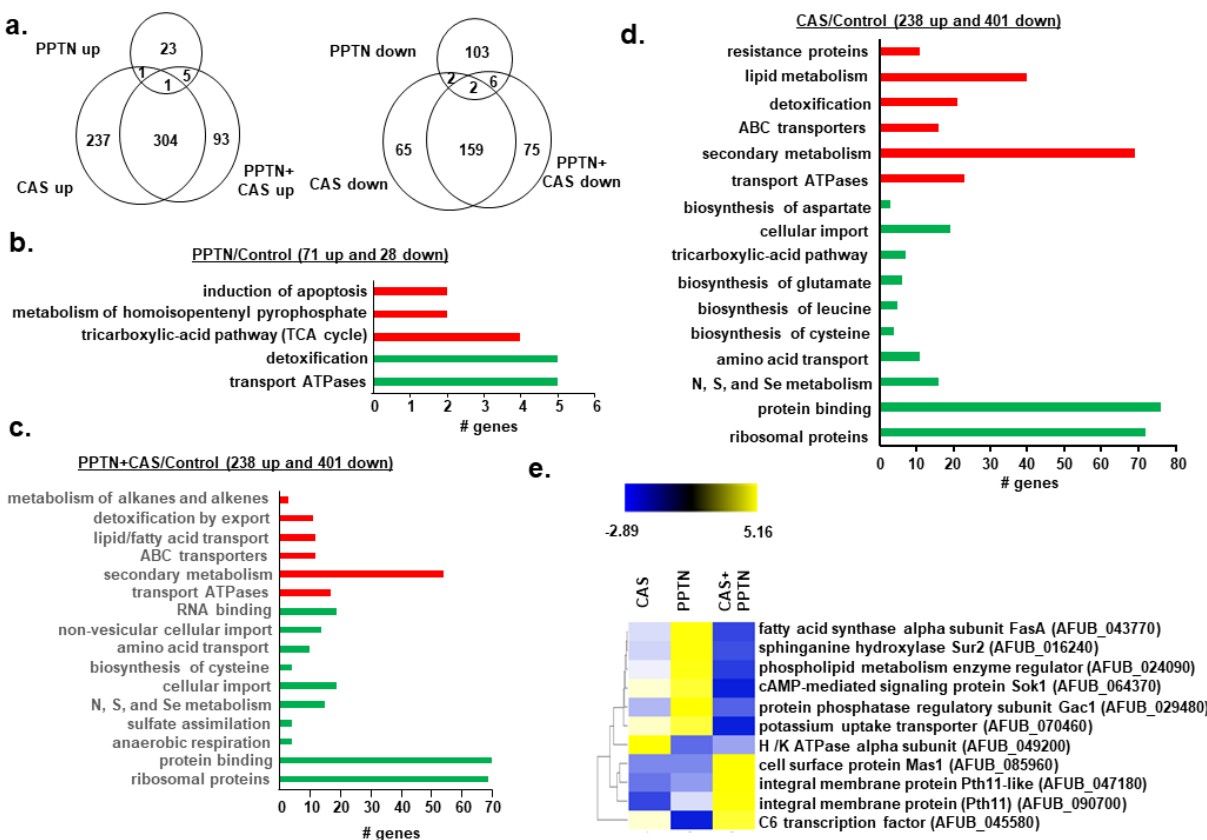

**FIG 6** Transcriptional profiling of *A. fumigatus* exposed to PPTN, CAS, or PPTN + CAS. (a) Venn diagram of upregulated and downregulated differentially expressed genes when *A. fumigatus* was exposed to PPTN, CAS, or PPTN + CAS. Fungifun categorization of differentially expressed genes when *A. fumigatus* was exposed to PPTN (b.), CAS (c.), and PPTN + CAS (d.). Heat map of genes visually selected that have a significant differential expression when *A. fumigatus* was exposed to PPTN, CAS, and PPTN + CAS. (e) Heat map for the RNAseq values of 11 selected genes. Hierarchical clustering was performed in MeV (https://webmev.tm4.org/), using Pearson correlation with complete linkage clustering.

## The combination of CAS with PPTN shows therapeutic potential in *in vitro* models

Given the profound effect CAS with PPTN had against diverse drug-resistant *A. fumigatus* strains, we assessed whether the compound combination had any toxic effects against mammalian cells. The cytotoxicity of CAS with PPTN or PPTN-NC was assayed using the human immortalized epithelial cell lineage A549 and liver immortalized HepG2 cells (Fig. 7a through d). Confluent cultures of A549 and HepG2 cells were exposed to increasing concentrations of PPTN with or without high concentrations of CAS for 24 h (A549) or 48 h (HepG2). Metabolic activity was assessed by XTT or Alamar blue assays, respectively (Fig. 7). In comparison to the control, when the cells were incubated with CAS, no significant toxicity to A549 cells was observed. With PPTN, higher concentrations (80 µM, 160 µM, and 320 µM) resulted in reduced metabolic activity of 25%, 88%, and 91%, respectively, relative to untreated controls (Fig. 7a). With PPTN-NC, higher concentrations (80 µM, 160 µM, and 320 µM) showed reduced toxicity with reduced activity of 25%, 35%, and 45%, respectively, relative to the untreated controls (Fig. 7b). When mammalian cytotoxicity was examined with compound combination in A549 epithelial cells, no enhanced toxicity was observed at PPTN or NC-PPTN 20 or 40 µM + CAS 100 µg/mL (Fig. 7a and b). By contrast, in A549 cells, a combination of PPTN or NC-PPTN 80, 160, or 320 µM + CAS 100 µg/mL resulted in a metabolic activity decrease of 25% and 80%, respectively (Fig. 7a and b). In HepG2 cells, PPTN 10, 20, 40, and 80 µM + CAS 50 µg/mL showed a metabolic decrease of 20%, 50%, 80%, and 90%,

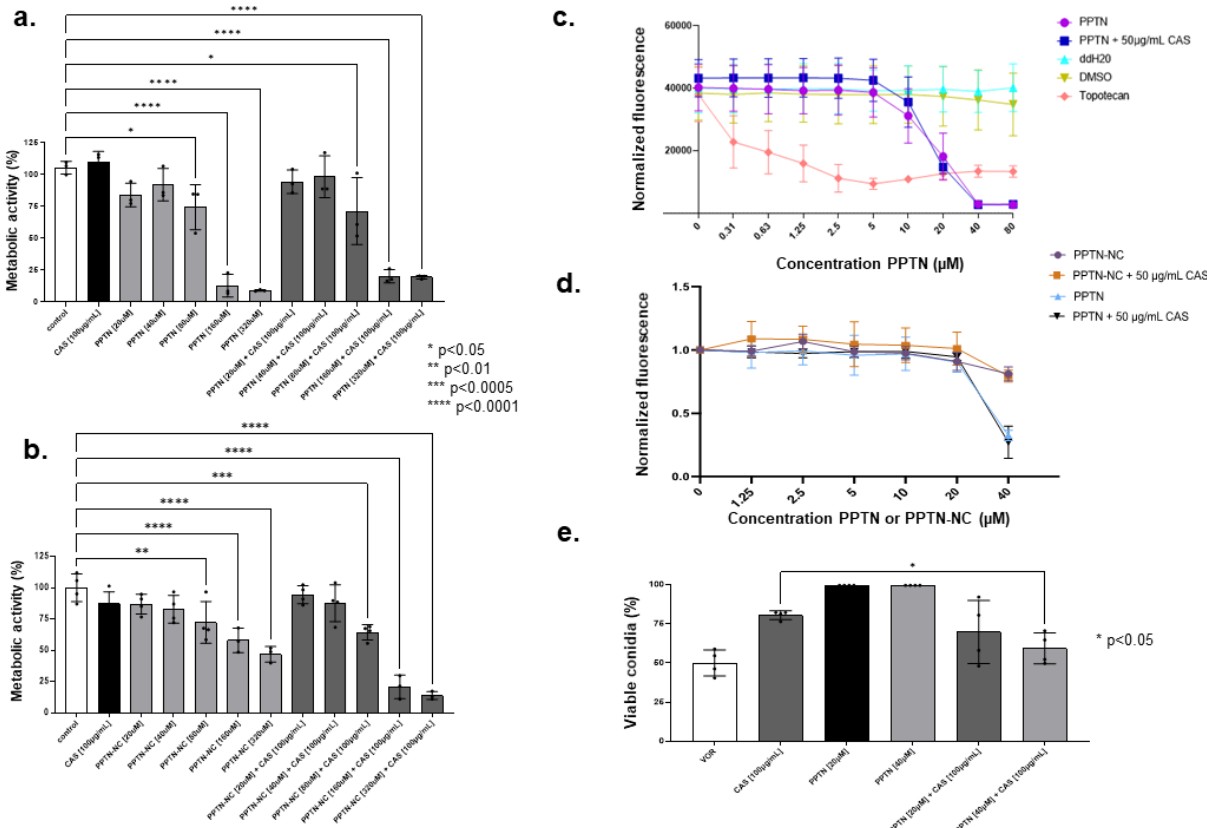

**FIG 7** The combination of PPTN + CAS is not cytotoxic to A549 cells and promotes the killing of *A. fumigatus* conidia in co-culture. (a and b) The human cells A549 were exposed to different concentrations of PPTN or PPTN-NC in the presence or absence of CAS (100 µg/mL). Untreated cells were used as a positive control. The graphic shows the value of nine technical replicates from three independent experiments ±SD (*n* = 3; two-way ANOVA, Tukey's post-test;*P* <0.05. ****P* < 0.0001. (b and c) HepG2 liver cells were treated with 50 µg/mL of CAS and 0–80 µM of PPTN or PPTN-NC alone, and in combination with 50 µg/mL of CAS, and after 48 h metabolic activity was evaluated by Alamar blue assay. Positive and negative controls were topotecan, DMSO, and double distilled (dd) water, respectively. The results represent the average of two independent experiments performed in technical duplicate. (e) A549 cells were challenged with *A. fumigatus* conidia at a multiplicity of infection of 1:10 in the absence or presence of different concentrations of PPTN and CAS. After 24 h of incubation, the media was removed and the cell suspension was plated in Sabouraud Dextrose Agar Media for 24 h at 37°C. The CFU percentage for each sample was calculated, and the results were plotted using GraphPad Prism (GraphPad Software, Inc., La Jolla, CA, USA). The graphic shows the value of nine technical replicates from three independent experiments ±SD (*n* = 3; two-way ANOVA, Tukey's post-test;*P* <0.05).

respectively (Fig. 7c). NC-PPTN +CAS 50 µg/mL was less toxic to the HepG2 cells with no decreased metabolic activity in 10 and 20 µM, but with 75% decreased metabolic activity in 40 µM (Fig. 7d). Notably, positive controls like DMSO or the anti-topoisomerase II agent topotecan showed very high toxicity to both cell lineages (Fig. 7a and c).

To assess the therapeutic potential of PPTN with CAS *in vitro,* we tested the ability of the compound combination to promote the killing of *A. fumigatus* conidia when infecting A549 pulmonary cells. When A549 cells were exposed to PPTN (20 or 40 µM) or CAS (100 µg/mL) alone, conidial killing reached 20%. Treatment with the VOR control (0.5 µg/mL) resulted in approximately 50% conidial killing. Finally, the combination of PPTN with CAS showed about 30% to 60% *A. fumigatus* conidial killing (Fig. 7e). Taken together, these data suggest that the combination of PPTN with CAS has minimal toxicity to mammalian cells and can enhance clearance of *A. fumigatus* infection in A549 pulmonary cells in *in vitro* assays.

## PPTN + CAS against other human fungal pathogens

Finally, to assess the spectrum of activity of PPTN and CAS, we investigated whether PPTN + CAS could increase the fungicidal activity against other human fungal pathogens, such as *Candida albicans*, *Candida auris*, *Candida glabrata*, *Candida krusei*, *Candida parapsilosis*, and *Cryptococcus neoformans*. Through dose-response assays, we observed that the MIC for PPTN in *Candida* species was >80 µM, and for *C. neoformans* was 10 µM. When examining the compound combination, complete inhibition of *C. albicans* metabolic activity (XTT) and survival (CFUs) shifted from a concentration of CAS at ~0.125 µg/mL in the absence of PPTN to a concentration of 0.015 µg/mL of CAS when 20 µM of PPTN was added, resulting in an eightfold reduction in the MIC of CAS (Fig. 8a). PPTN also suppressed CAS-resistance of *C. albicans* CAS-resistant clinical isolates, resulting in significant reductions in metabolic activity with the compound combination relative to individual drug treatments (Fig. 8b). For the emerging pathogen *C. auris,* PPTN with CAS significantly inhibited *C. auris* metabolic activity in diverse clinical isolates relative to individual drug treatments (Fig. 8c). For one of the clinical isolates, 467/2015, PPTN with CAS resulted in a 100% reduction in metabolic activity and survival, potentiating caspofungin activity by at least eightfold (Fig. 8d). We also observed significant reductions in the metabolic activity of *C. glabrata*, *C. krusei*, and *C. parapsilosis* when exposed to PPTN with CAS relative to individual drug treatments (Fig. 9a). CAS lacks activity against *C. neoformans* due to intrinsic resistance (Johnson and Perfect, 2003) such that only CAS concentrations above 16 µg/mL can inhibit *C. neoformans* metabolic activity (as determined by XTT) and survival (as determined by colony-forming units [CFUs]) (Fig. 9b). Intriguingly, PPTN at 2.5 µM (0.25× MIC)+ CAS had increased fungicidal activity at 16 µg/mL of CAS, resulting in complete inhibition of *C. neoformans* metabolic activity and growth (Fig. 9c). Taken together, these results indicate that PPTN + CAS have increased fungicidal activity against different human fungal pathogens.

## DISCUSSION

The chemical armamentarium to treat aspergillosis is very limited. Decades of poor progress in drug development have resulted in only a few antifungal agents in different stages of development, with some in clinical trials or under approval from regulatory agencies (58, 59). Combined with the few treatment options to treat IPA, the emergence of azole resistance among *Aspergillus* strains is also a growing concern (60–62), highlighting the urgent need to develop new therapeutic options. The development of new antifungal drugs is hampered because this is a costly and time-consuming process. This can potentially be circumvented by repurposing drugs that are already on the market but are licensed for use in other types of diseases. Thus, the screening of chemical libraries to identify preexisting drugs with new antifungal activity has been explored, representing an interesting, cheaper, and faster approach to discover new antifungal compounds (35–37, 63–65). Using this approach, our group identified 12 compounds that synergize with the echinocandin caspofungin (CAS) against *A. fumigatus (43)*. Against *A. fumigatus,* echinocandins are primarily fungistatic, and this species is intrinsically tolerant to echinocandins exhibiting residual growth even at high concentrations of the drug (66). Therefore, the conversion of CAS to a fungicidal drug against *A. fumigatus* is of great interest. Here, we describe in more detail the microbiological and molecular effects of combining PPTN with CAS. PPTN is a compound that antagonizes GPCR belonging to a family of receptors named P2Y receptors (44).

PPTN and its modified version NC-PPTN (that has no activity in mammalian cells) are not only synergistic with CAS but also with VOR and AMPHO, showing the broad spectrum of action of PPTN when it is combined with other commercial antifungal drugs. In addition, the growth of VOR- and CAS-resistant *A. fumigatus* clinical isolates was also inhibited upon the addition of PPTN with CAS, demonstrating the efficacy of this combination against drug-resistant strains. Furthermore, *A. fumigatus* can form robust biofilms within the infection environment (67, 68) and these structures have great clinical importance since biofilm formation is related to increased resistance to

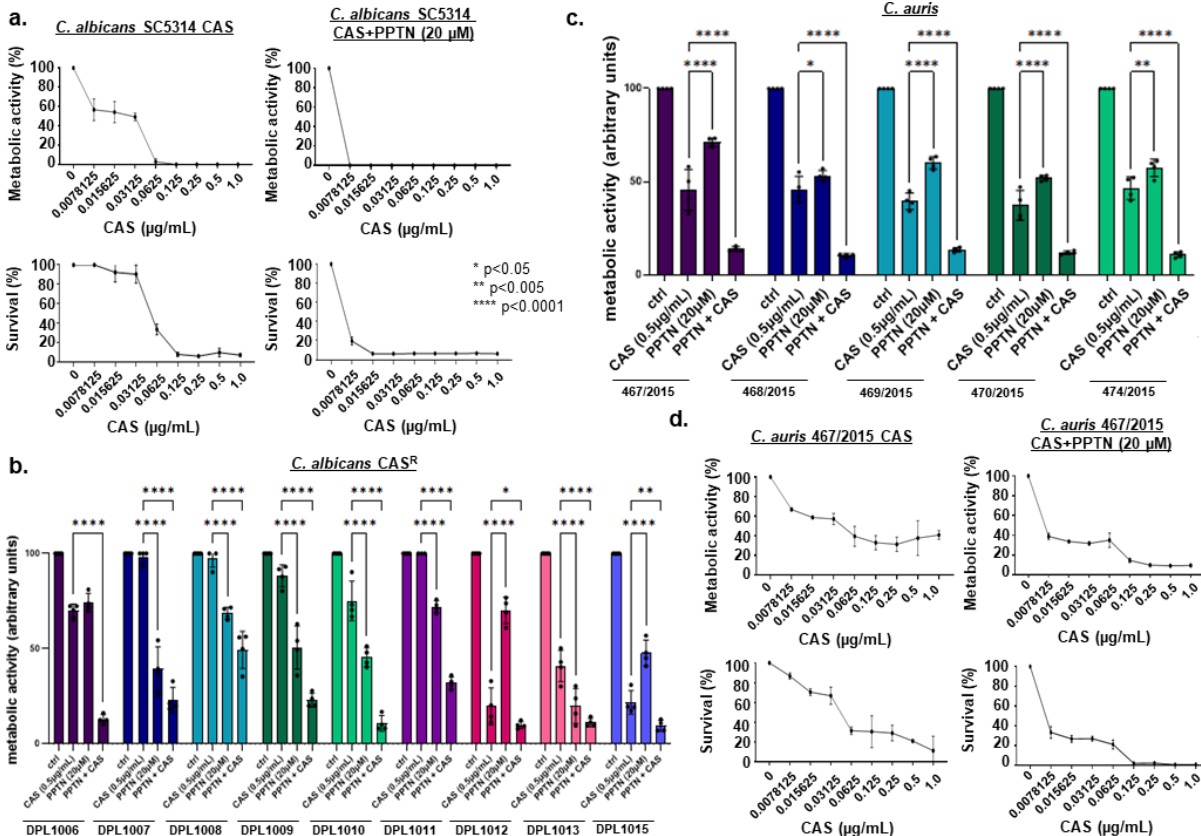

**FIG 8** PPTN synergizes with CAS in *C. albicans* and *C. auris*. (a) Metabolic activity expressed by XTT of *C. albicans* grown for 48 h in the absence or presence of CAS (0 to 1 µg/mL) or PPTN (20 µM) + CAS (0–1 µg/mL). Percentage of survival expressed as colony-forming units/mL of *C. albicans* cells grown for 48 h in the absence or presence of CAS (0–1 µg/mL) or PPTN (20 µM)+ CAS (0–1 µg/mL). Experiments were carried out in three replicates for each sample and three independent experiments ($n = 3$). (b) Metabolic activity expressed by XTT of *C. albicans* caspofungin-resistant (CAS[R]) grown for 48 h in the absence or presence of CAS (0.5 µg/mL), PPTN (20 µM), or PPTN (20 µM) + CAS (0.5 µg/mL). (c) Metabolic activity expressed by XTT of *C. auris* grown for 48 h in the absence or presence of CAS (0.5 µg/mL), PPTN (20 µM), or PPTN (20 µM) + CAS (0.5 µg/mL). (d) Percentage of survival expressed as colony-forming units/mL of *C. auris* cells grown for 48 h in the absence or presence of CAS (0–1 µg/mL) or PPTN (20 µM) + CAS (0–1 µg/mL). Experiments were carried out in three independent replicates ($n = 3$), with three technical replicates, and analyzed by two-way ANOVA followed by Dunnett's multiple comparison test (*$P < 0.05$ and ****$P < 0.0001$).

antifungal drugs (69–71). CAS in combination with PPTN was able to reduce up to 51% of the metabolic activity of a mature fungal biofilm, suggesting that the combination has potential antibiofilm activity against *A. fumigatus*. Importantly, PPTN-NC was shown to be less toxic to mammalian cells, which suggests the possibility of developing efficient PPTN derivatives that are toxic to the fungus but not to the host.

PPTN converts CAS into a fungicidal drug. Death triggered by antifungal drugs may be the result of alterations in different cellular pathways. For instance, the activation of fungal metacaspases culminates with events leading to cell death by apoptosis-like programmed cell death (49, 72). Fragmentation of the fungal mitochondrial tubular network has also been described as a marker for cell death (50, 51). In contrast to an apoptosis-like programmed cell death pathway, increased cell death due to external factors (a necrotic-like pathway) was observed for PPTN with CAS. Interestingly, HIP performed in *S. cerevisiae* identified several heterozygous deletion mutants for genes involved in chromatin remodeling as hypersensitive to PPTN, and synergy between PPTN and trichostatin A (a histone deacetylase inhibitor) and 5-AZA (an inhibitor of DNA methyltransferase) was also observed in *A. fumigatus*. While these results are intriguing, it remains to be determined how the combination of PPTN and CAS affects permeability and chromatin remodeling or other biological processes such as DNA repair.

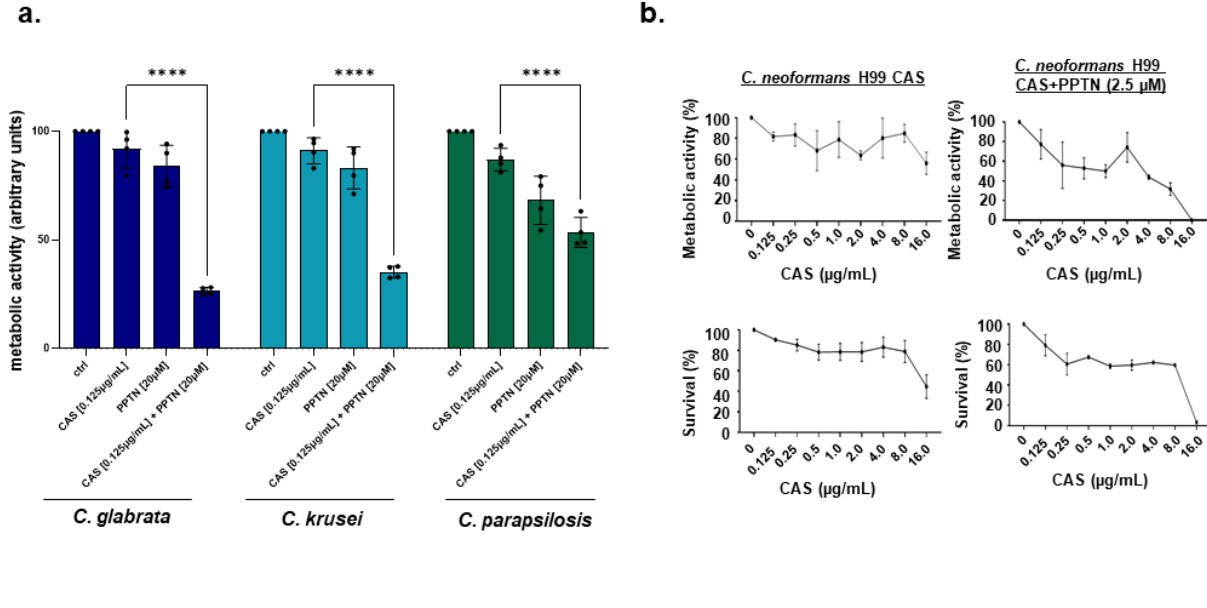

**** p<0.0001

FIG 9 PPTN synergizes with CAS in *C. glabrata*, *C. krusei*, and *C. parapsilosis,* but not *C. neoformans*. (a) Metabolic activity expressed by XTT of *C. glabrata*, *C. krusei*, and *C. parapsilosis* grown for 48 h in the absence or presence of CAS (0.125 µg/mL) or PPTN (20 µM ) + CAS (0.125 µg/mL). Experiments were carried out in three replicates for each sample and three independent experiments (*n* = 3). (b) Metabolic activity expressed by XTT of *C. neoformans* grown for 48 h in the absence or presence of CAS (0–32 µg/mL), PPTN (20 µM), or PPTN (20 µM) + CAS (0–32 µg/mL). Percentage of survival expressed as colony-forming units/mL of *C. neoformans* cells grown for 48 h in the absence or presence of CAS (0–32 µg/mL) or PPTN (20 µM ) + CAS (0–32 µg/mL). Experiments were carried out in three independent replicates (*n* = 3), with three technical replicates, and analyzed by two-way ANOVA followed by Dunnett's multiple comparison test (*$P$ < 0.05 and ****$P$ < 0.0001).

We assessed the PPTN cytotoxicity in two human immortalized cell lineages (A549 pulmonary epithelial cells and HepG2 hepatic cells) exposed to increasing concentrations of PPTN with or without CAS. While differences were noted between the two cell lines, likely due to intrinsic metabolic or other differences in both cell lineages, minimal toxicity was observed at compound concentrations with antifungal activity. This suggests that although PPTN has some toxicity, it is still able to promote, at least partially, the clearance of *A. fumigatus* conidial cells in co-culture.

Importantly, we also observed that the combination of PPTN and CAS significantly reduced the metabolic activity of different fungal pathogens such as *C. albicans*, CAS-resistant *C. albicans*, *C. auris, C. glabrata*, *C. krusei*, and *C. parapsilosis*, showing that the antifungal properties of the drug combination are effective against several fungal pathogens. However, PPTN can only interact with high CAS concentrations (16 µg/mL) against *C. neoformans*. This work provides, for the first time, evidence that PPTN, an antagonist of the GPCRs belonging to the P2Y receptors, can synergize with CAS, controlling the growth of several fungal pathogens. Further work will focus on better understanding the mode of action of PPTN against pathogenic fungal species.

## MATERIALS AND METHODS

### Strains, media, and cultivation method

All fungal species used in this work (*A. fumigatus. Candida* spp. and *C. neoformans*) are listed in Table S1; https://doi.org/10.6084/m9.figshare.28152551). The *A. fumigatus* strains were grown and maintained in solid or liquid minimal medium at 37°C (liquid MM: 1% (wt/vol) glucose, 50 mL of 20× salt solution, trace elements, 2% (wt/vol), pH 6.5; solid MM contains the same elements but added of 2% agar). Solutions of trace elements and

salt solutions are described by Käfer (1977). *Candida* spp. and *C. neoformans* strains were grown and maintained in YPD (1% yeast extract, 2% peptone, and 2% glucose).

## Minimal inhibitory concentration

For the initial assays, the PPTN and NC-PPTN compounds were kindly offered by the Structural Genomics Consortium (SGC; https://www.thesgc.org/) and solubilized in DMSO (Fisher Scientific, Loughborough, UK) at 10 mM. For additional experiments, the PPTN was acquired from Sigma-Aldrich (SML1809) and it was used after solubilizing in DMSO or PBS as described. The MIC analysis of *A. fumigatus* was performed according to the M38-A2 protocol of the Clinical and Laboratory Standards Institute (73) with modifications. Briefly, the assay was performed using 200 µL of MM, pH 6.5 supplemented with PPTN (160 to 2.5 µM) and $1 \times 10^4$ conidia/mL. Plates were incubated at 37°C without shaking for 48 h. The MIC analysis of yeast fungal pathogens was performed according to the M27-A3 protocol using the RPMI-1640 medium containing PPTN (160 to 2.5 µM). The yeast plates were incubated at 30°C for 48 h (*Candida* spp.) or 72 h (*C. neoformans*). Wells containing only medium and PBS or DMSO were used as a control. As the PPTN did not inhibit the growth of any strain, the MIC value (the lowest concentration of PPTN that visually inhibited 100% of fungal growth) was not defined. All experiments were done in triplicate.

## Definition of fungicidal activity of PPTN combined with subinhibitory concentrations of antifungal drugs

A total of $1 \times 10^4$ conidia/mL of *A. fumigatus* wild-type strain was inoculated in 96-well plates, each well containing 200 µL of liquid MM added of PPTN (20 µM) in the presence or absence of increasing concentration of CAS. Plates were incubated for 48 h at 37°C without shaking. Then, the equivalent of 100 conidia was plated in solid MM and incubated at 37°C for another 36 h. Wells containing only medium and DMSO were used as controls. The number of viable colonies was determined by counting the number of CFUs and expressed in comparison with the control (no germinated and untreated fresh conidia, which was considered 100% survival). Results are expressed as means and standard deviations (SD) from three independent experiments.

## Interaction between PPTN and commercial antifungals against *A. fumigatus* and yeasts

The interaction of PPTN with antifungal drugs against *A. fumigatus* was assayed using a checkerboard microdilution method using Alamar blue (Life Technologies) according to Yamaguchi and colleagues (2002). This methodology shows the metabolic activity of each fungal species during the exposure of the strain to specific drug combinations. Briefly, conidia of *A. fumigatus* ($2.5 \times 10^4$ cells/mL) were *resuspended in* liquid MM added 10% Alamar blue plus a specific concentration of each drug. A total of 100 µL was dispensed per well of a 96-well plate. The drugs used were PPTN (0–80 µM), caspofungin (CAS; 0–8 µg/mL), amphotericin B (0–8µg/mL), voriconazole (4 µg/mL), and ibrexafungerp (IBX; 0–1 µg/mL). As positive controls, the drugs were replaced by the same volume of the medium, and as the negative control, wells were filled in the same medium but without conidia. Plates were incubated for 48 h at 37°C without shaking in the dark, and the results were read spectrophotometrically by fluorescence (570 nm excitation– 590 nm emission) in a microplate reader (SpectraMax Paradigm Multi-Mode Microplate Reader; Molecular Devices). The interaction score between PPTN and commercial antifungal compounds was assessed with SynergyFinder2.0 (https://synergyfinder.fimm.fi), a web application for the analysis of drug combination responses. The interaction between different drugs was classified based on the synergy score, where values lower than −10 were considered antagonistic, values from −10 to 10 were considered additive, and values higher than 10 were considered synergistic.

The effect of the combination between PPTN and CAS against yeast fungal pathogens was assessed by metabolic activity using XTT and the CFUs. A total of $10^4$ cells from *C.*

*neoformans*, *C. albicans*, and *C. auris* was inoculated in RPMI-1640 supplemented with CAS (0–32 µg/mL for *C. neoformans* and 0–1 µg/mL for *Candida* spp.) or CAS combined with PPTN (20 µM for *Candida* spp. or 2.5 µM for *C. neoformans*). After 48 h of incubation, the metabolic activity of viable cells was revealed using the XTT assay (26). For CFU determination of the yeast cells, the same experimental design was used, but after 48 h, the cells were plated on YPD and let to grow at 30°C for 24–48 h to determine the survival percentage. The results are the average of three repetitions and are expressed as average ± standard deviation.

## Fluorescence microscopy

Conidia ($10^5$) of specific strains were inoculated on coverslips in 4 mL of MM and let to grow 16 h at 30°C. Adherent germlings were left untreated or treated with PPTN, CAS, or a combination of both drugs for different periods of time as indicated. Staining was performed as follows: (i) MM added of 50 µg/mL propidium iodide for 5 min (Sigma-Aldrich); (ii) MM added of 20 ug/mL of Hoechst 33342 dye for 10 min (Molecular Probes, Eugene, OR, USA). Furthermore, the coverslips were rinsed with PBS (140 mM NaCl, 2 mM KCl, 10 mM NaHPO4, 1.8 mM KH2 PO4, pH 7.4). Slides were visualized on the Observer Z1 fluorescence microscope using a 100 oil immersion lens objective. Differential interference contrast (DIC) images and fluorescent images were captured with an AxioCam camera (Carl Zeiss) and processed using AxioVision software (version 4.8). In each experiment, at least 50 germlings were analyzed. For GFP, the wavelength excitation was 450 to 490 nm, and the emission wavelength was 500–550 nm. For RFP and PI, the wavelength excitation was 572/25 nm and the emission wavelength was 629/62 nm. For Hoechst (4,6-diamidino-2-phenylindole) staining, the excitation wavelength was 365 nm and the emission wavelength was 420 to 470 nm.

## mito::GFP strain construction

The mito::GFP strain was constructed through the gene replacement cassettes approach using "in vivo" recombination in *S. cerevisiae* as previously described in reference (74). Briefly, about 1.0 kb from the 5′-UTR of the pyrG gene was amplified (primers P1/P2) from *A. fumigatus* genomic DNA (gDNA). The gpdA sequence was amplified from the gDNA of the *A. nidulans* AGB655 strain (primers P3/P4). The 5′-UTR of the citrate synthase gene was PCR amplified from *A. niger* gDNA (primers P5/P6), the GFP sequence was PCR amplified from pMCB17apx plasmid (provided by Vladimir P. Efimov; primers P7/P8) and the pyrG gene was amplified from the *A. fumigatus* gDNA (primers P9/P10). The cassette was generated by transforming each fragment along with the plasmid pRS426 linearized with *Bam*HI/*Eco*RI into the *S. cerevisiae* strain SC9721 using the lithium acetate method (75). The DNA from the transformants was extracted by the method described in reference (76), and PCR was run to confirm the correct construction. The whole cassette was then PCR-amplified from *S. cerevisiae* DNA (primers P1/P9) and used to transform *A. fumigatus* strain (77). *A. fumigatus* candidates were selected and purified through three rounds of growth on plates. The gDNA of the mutants was extracted, and the construction insertion was confirmed by PCR (primers P11/P6). To note, all DNA fragments were PCR-amplified with TaKaRa Ex Taq DNA Polymerase (Clontech USA) and primers P1 and P9 contained a short homologous sequence to the MCS of the plasmid pRS426. A list of all primer pairs is shown in Table S3; https://doi.org/10.6084/m9.figshare.28152551).

## Haploinsufficiency profiling

We utilized a heterozygous deletion library consisting of 968 uniquely barcoded diploid *S. cerevisiae* strains that were constructed in a *pdr1Δ/Δ pdr3Δ/Δ snq2Δ/Δ* (or *3Δ*) compound-sensitized background (54). Briefly, 96-well plates were prepared with 100 µL containing PPTN in a twofold dilution series (six technical replicates per concentration), alongside six wells containing a DMSO solvent control. The 3Δ library (stored in aliquots at −80°C) was thawed on ice and diluted 1/100 in YPD (e.g., 100 µL in 10 mL). 100 µL of

this diluted pool was then added to each well of the 96-well plate (highest concentration of PPTN tested = 25 µM). Plates were wrapped in foil and incubated at 30°C for 48 h. At 48 h, the highest concentration of PPTN for which there was sufficient growth to harvest (at least 75% of the DMSO control by OD$_{600}$ measurement) was selected for HIP (6.25 µM), along with the DMSO controls. The pooled cultures were transferred to microcentrifuge tubes (two technical replicates per microcentrifuge tube, to ensure adequate gDNA isolation), centrifuged at 16,000 × $g$ for 1 min, and the supernatant was discarded. Pellets were digested with 15 units of Zymolyase (US Biological Z1004) in 500 µL Zymolase buffer (1 M sorbitol, 10 mM sodium EDTA, 14 mM beta-mercaptoe-thanol) at 37°C for 1 h. gDNA extraction was performed using the PureLink Genomic DNA Extraction kit, as per the manufacturer's instructions (Invitrogen), and gDNA values across all samples were standardized by quantifying gDNA with the Quant-iT PicoGreen dsDNA Assay Kit (Thermo Scientific). PCR (30 cycles) was then used to amplify the strain-specific barcodes present in each gDNA sample, with a universal reverse primer and a unique indexed forward primer ([54]. The 267 bp amplicons were then pooled together (10 µL from each reaction), run on an agarose gel, and gel-purified using a QIAquick Gel Extraction kit following the manufacturer's instructions (Qiagen). The resulting pooled amplicons were sequenced on an Illumina NextSeq500 instrument (Mid-Output, V2 Chemistry). BEAN-counter was used to demultiplex the resulting fastq file and return strain-specific barcode abundance for each gDNA sample normalized to control treatments (78), thus producing chemical-genetic scores to reflect strain-specific abundance in the presence of compound treatments. The resulting chemical-genetic scores were plotted and visualized using R, using 6× the median absolute deviation (MAD) as a stringent cut-off for significantly depleted strains.

## Growth curves with *S. cerevisiae*

Significantly depleted strains from HIP with PPTN were validated as hypersensitive by analysis of growth curves for the *3Δ* parent of the HIP pool (ScLC5233) and individual heterozygous mutants from the *3Δ* pool. This entailed growing *S. cerevisiae* strains in the absence (−) and presence (+) of PPTN (three technical replicates per strain and per compound treatment) at 30°C and calculating the area under the curve (AUC) for the (−) and (+) curves based on OD$_{600}$ values (taken every hour for 48 h). A ratio (AUC(−)/AUC(+)) was calculated for each strain, and the heterozygous mutants were compared to the 3Δ parent strain to determine if hypersensitivity was observed.

## RNA purification and preparation for RNA-seq

A total of $10^6$ spores/mL of *A. fumigatus* wild-type strain (CEA17) was inoculated in 50 mL of liquid MM at 37°C for 16 h under agitation. Then, the mycelia were exposed to PPTN (20 µM), CAS (0.25 µg/mL), or a combination of PPTN plus CAS (20 µM + 0.25 µg/mL), respectively for 1 h at 37°C. Total RNA was extracted by the TRIzol method, treated with RQ1 RNase-free DNase I (Promega), and purified using the RNAeasy kit (Qiagen) according to the manufacturer's instructions. The total RNA was quantified using a NanoDrop and its integrity was analyzed using an Agilent 2100 Bioanalyzer. All RNA had a minimum RNA integrity number (RIN) value of 8.0. For RNA-sequencing, the cDNA libraries were constructed using the TruSeq Total RNA with Ribo Zero (Illumina, San Diego, CA, USA). From 0.1 to 1 µg of total RNA, the ribosomal RNA was depleted and the remaining RNA was purified, fragmented, and prepared for complementary DNA (cDNA) synthesis, according to the manufacturer's recommendations. The libraries were validated following the Library Quantitative PCR (qPCR) Quantification Guide (Illumina). Following, the RNA-seq was carried out by paired-end sequencing on the Illumina NextSeq 500 Sequencing System using NextSeq High Output (2 × 150) kit, according to the manufacturer's recommendations.

## A549 cytotoxicity assay

The cytotoxicity of PPTN and PPTN-NC was assessed in A549 human lung cancer cells using the XTT assay. Cells ($2 \times 10^5$ cells/well) were seeded in 96-well tissue plates and incubated in Dulbecco's modified Eagle medium (DMEM) culture medium. After 24 h of incubation, the cells were treated with PPTN (20, 40, 80, 160, and 320 µM), CAS (100 µg/mL), or in different PPTN +CAS combinations. After 24 h incubation, the metabolic activity was assessed using the XTT kit (Roche Applied Science) according to the manufacturer's instructions. Formazan formation was quantified spectrophotometrically at 450 nm (reference wavelength 620 nm) using a microplate reader. The experiment was performed in three replicates. Viability was calculated using the background-corrected absorbance as follows: Metabolic activity (%) =absorbance value of experiment well/absorbance value of control well $\times$ 100.

## HepG2 cytotoxicity assay

For the PPTN assay, HepG2 cells were seeded at $1.25 \times 10^5$ cells/mL into 96-well plates (Starstedt) in a total volume of 100 µL of RPMI supplemented with 10% FBS. Plated cells were then grown overnight at 37°C in the presence of 5% $CO_2$. Compounds, dissolved in ddH$_2$O, with appropriate controls, were then twofold diluted into source plates, and 100 µL of each concentration was subsequently added to cells. Compounds dissolved in DMSO were added using a Tecan D300e compound dispenser at the indicated final concentrations. Growth was then continued for an additional 48 h. Following 48 h of growth, 50 µL of a 1:4 dilution of Alamar Blue viability reagent (Invitrogen) was added to each well, and plates were incubated for an additional 3 h at 37°C in the presence of 5% $CO_2$. Fluorescence measurements (535 nm excitation/595 nm emission) were performed using a Tecan Spark multimode microplate reader to quantify metabolic activity. Fluorescence measurements were corrected for background from an average of medium controls across all plates, and normalized relative metabolic activity was calculated by dividing an equal volume titration of relevant average solvent control and propagating standard deviation error. All experiments were performed in technical triplicate and biological duplicate.

For the PPTN-NC assay, HepG2 cells in RPMI supplemented with 10% FBS were seeded at $5 \times 10^4$ cells/mL into 384 well white clear bottom plates (Starstedt) for a total volume of 40 µL. Plated cells were then grown overnight at 37°C in the presence of 5% $CO_2$. Compounds, dissolved in DMSO, were then twofold diluted using the TECAN D300e digital dispenser starting at 40 µM and growth was continued for an additional 48 h. Following growth, 10 µL of a 1:4 dilution of Alamar Blue viability reagent (Invitrogen) was added to each well, and plates were incubated for an additional 3 h at 37°C in the presence of 5% $CO_2$. Fluorescence measurements (535 nm excitation/595 nm emission) were performed using a Tecan Spark multimode microplate reader to quantify metabolic activity. Fluorescence measurements were corrected for background from an average of medium controls across all plates. Data plotted in GraphPad Prism Version 10.3.1 (464) represent results from one of the two biological replicates, performed in technical quadruplicate, that yielded similar results.

## Killing assay

The human lung cancer cell A549 was cultured using DMEM (Thermo Fisher Scientific, Paisley, UK) supplemented with 10% fetal bovine serum (FBS) and 1% penicillin–streptomycin (Sigma-Aldrich, Gillingham, UK). A total of $10^6$ cells/mL was seeded in 24-well plates (Corning) for 24 h at 37°C, 5% $CO2._2$ The cells were treated with PPTN (20 and 40 µM), CAS (100 µg/mL), or in different combinations between them and challenged with *A. fumigatus* conidia at a multiplicity of infection of 1:10. After 24 h of incubation in 5% $CO_2$, the culture media was removed, and the wells were washed with 2 mL of sterile water. The A549 cells were lysed by the addition of 2 mL of cold sterile water and the cell suspension was collected. This suspension was then diluted 1:1,000, and 100 µL was plated on Sabouraud Dextrose Agar Media (SAB). After 24 h incubation at

37°C, the number of CFUs was determined. To check the inoculum viability, 50 µL of the original inoculum adjusted to $10^3$ /mL was also plated on SAB agar to correct CFU counts. The CFU percentage for each sample was calculated and the results were plotted using GraphPad Prism (GraphPad Software, Inc., La Jolla, CA, USA). A $P$ value ≤ 0.001 was considered significant.

## Statistical analysis

Grouped column plots with standard deviation error bars were used for representations of data. For comparisons with data from wild-type or control conditions, we performed one-tailed, paired $t$ tests or one-way analysis of variance (ANOVA) or a Student's t-test (when analyzing only two samples). All statistical analyses and graphics building were performed using GraphPad Prism 5.00 (GraphPad Software). A $P$ value ≤ 0.05 was considered statistically significant.

## ACKNOWLEDGMENTS

We thank the Fundação de Amparo à Pesquisa do Estado de São Paulo (FAPESP) grant numbers 2021/04977-5 (to G.H.G.), 2023/07785-5 (to T.F.D.R.), and 2023/00206-0 (to E.D.), the Conselho Nacional de Desenvolvimento Científico e Tecnológico (CNPq), and Fundação Coordenação de Aperfeiçoamento do Pessoal do Ensino Superior (CAPES) grant number 405934/2022-0 (The National Institute of Science and Technology INCT Funvir) and CNPq 301058/2019-9 (to G.H.G.), both from Brazil, and the National Institutes of Health/National Institute of Allergy and Infectious Diseases grant R01AI153356 (to G.H.G.), from the US. We thank Marcela Savoldi and Lívia Alfaya for their technical assistance.

This work was also funded by the Joint Canada-Israel Health Research Program, jointly supported by the Azrieli Foundation, Canada's International Development Research Centre, Canadian Institutes of Health Research, and the Israel Science Foundation (G.H.G. and L.E.C.). The views expressed herein do not necessarily represent those of IDRC or its Board of Governors. LEC is also supported by the Canadian Institutes of Health Research (CIHR) Foundation grant (FDN-154288), and is a Canada Research Chair (Tier 1) in Microbial Genomics & Infectious Disease and co-director of the CIFAR Fungal Kingdom: Threats & Opportunities program.

## AUTHOR AFFILIATIONS

[1]Faculdade de Ciências Farmacêuticas de Ribeirão Preto, Universidade de São Paulo, São Paulo, Brazil
[2]National Institute of Science and Technology in Human Pathogenic Fungi, São Paulo, Brazil
[3]Centro de Biociências, Universidade Federal do Rio Grande do Norte, Natal, Brazil
[4]Structural Genomics Consortium, University of Toronto, Toronto, Ontario, Canada
[5]Department of Molecular Genetics, University of Toronto, Toronto, Ontario, Canada

## AUTHOR ORCIDs

Thaila Fernanda dos Reis http://orcid.org/0000-0002-7776-977X
Bradley Laflamme http://orcid.org/0000-0001-8220-8128
Leah E. Cowen https://orcid.org/0000-0001-5797-0110
Gustavo H. Goldman http://orcid.org/0000-0002-2986-350X

## FUNDING

| Funder | Grant(s) | Author(s) |
| --- | --- | --- |
| Fundação de Amparo à Pesquisa do Estado de São Paulo (FAPESP) | 2021/04977-5 | Gustavo H. Goldman |

| Funder | Grant(s) | Author(s) |
|---|---|---|
| Fundação de Amparo à Pesquisa do Estado de São Paulo (FAPESP) | 2023/00206-0 | Endrews Delbaje |
| Conselho Nacional de Desenvolvimento Científico e Tecnológico (CNPq) | 405934/2022-0 | Gustavo H. Goldman |
| Conselho Nacional de Desenvolvimento Científico e Tecnológico (CNPq) | 301058/2019-9 | Gustavo H. Goldman |
| HHS | NIH | National Institute of Allergy and Infectious Diseases (NIAID) | R01AI153356 | Gustavo H. Goldman |
| Canadian Government | Canadian Institutes of Health Research (CIHR) | FDN-154288 | Leah E. Cowen<br>Gustavo H. Goldman |

## DATA AVAILABILITY

All the data are available as supplementary tables and figures. The RNA-seq data are available in NCBI's database under the BioProject ID PRJNA1165814.

## ADDITIONAL FILES

The following material is available online.

### Open Peer Review

**PEER REVIEW HISTORY (review-history.pdf).** An accounting of the reviewer comments and feedback.

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
