## [Reviewer comments · Microbiology Spectrum]

Microbiology Spectrum

The GPCR antagonist PPTN synergizes with caspofungin providing increased fungicidal activity against *Aspergillus fumigatus*

Thaila dos Reis, Endrews Delbaje, Camila Pinzan, Rafael Bastos, Suzanne Ackloo, Sara Fallah, Bradley Laflamme, Nicole Robbins, Leah Cowen, and Gustavo Goldman

Corresponding Author(s): Gustavo Goldman, Universidade de Sao Paulo

Review Timeline:

Submission Date:	December 18, 2024
Editorial Decision:	January 6, 2025
Revision Received:	January 7, 2025
Accepted:	January 31, 2025

Editor: James Konopka

Reviewer(s): Disclosure of reviewer identity is with reference to reviewer comments included in decision letter(s). The following individuals involved in review of your submission have agreed to reveal their identity: W. Scott Moye-Rowley (Reviewer #3)

Transaction Report:

DOI: <https://doi.org/10.1128/spectrum.03318-24>

Re: Spectrum03318-24 (The GPCR antagonist PPTN improves caspofungin fungicidal activity against *Aspergillus fumigatus*)

Dear Dr. Gustavo H. Goldman:

Thank you for the privilege of reviewing your work. Below you will find my comments, instructions from the Spectrum editorial office, and the reviewer comments.

The reviewers thought that the topic of your manuscript was interesting. However, Reviewer 1 had some concerns about how some of the conclusions were stated and how some of the assays were carried out.

Revision Guidelines

Sincerely,
James Konopka
Editor
Microbiology Spectrum

Reviewer #1 (Comments for the Author):

This manuscript describes the characterization of a compound that was previously reported by the investigative group to decrease the MEC of caspofungin when used in combination against *A. fumigatus*. Here, they report on the activity of this compound, alone and in combination with CAS and other antifungals, against a larger selection of *Aspergillus* and other fungal

isolates. The authors also provide cell biological and transcriptional studies to explore MOA of the compound and provide data regarding toxicity. In general, the studies are interesting and carry impact for the field. However, there are multiple issues with conclusions not being supported by the data and additional clarity of experimental approaches that are needed.

An overall issue that the authors need to consider is that caspofungin treatment may actually be potentiating the activity of PPTN, and not the other way around as they conclude throughout the manuscript and in the title. There is not much evidence provided that this is not, in fact, the case. PPTN does have some intrinsic antifungal activity as the authors report that it has measurable MICs against some *Aspergillus* isolates and against other pathogenic fungi. Also, PPTN showed synergy/additive effects with all of the antifungals used. The triazoles and polyenes caused weakened membranes. It would make more sense that each of these antifungals allow accumulation of PPTN in the fungal cell to enhance the antifungal effect of PPTN itself. Therefore, the antifungals potentiate PPTN. What evidence truly shows that PPTN potentiates the antifungals?

It would've been extremely helpful for the authors to have included line numbers. I have tried to utilize the appropriate page numbers below to assist the authors in locating the pertinent areas of the manuscript.

1. For Fig 1b: This data should not be expressed as "% of viable conidia". Due to the hyphal growth that occurs and the way these hyphae break apart into an unpredictable number of viable CFUs during removal from the 96-well plates and subsequent plating for calculations, there is no way to conclude that these values are predictive of the viability of conidia. This should just be expressed as CFU relative to untreated controls.
2. For Fig 1C: The authors need to clarify their wording related to how they describe this data. On Page 6, the authors state that "...the fungal cells were metabolically active up to 0.5 ug/ml (of CAS)". There are a wealth of publications showing there is no 100% MIC for CAS against *Aspergillus* species and, therefore, the cells are certainly metabolically active at this drug concentration. The inability to detect "metabolic activity" here is just a limitation of the assay.
3. For Table 1: The data do not support the conclusions, as stated. The authors state on Page 7, that "These results suggest that the combination of PPTN with CAS can overcome both CAS and VOR resistance...". However, the authors do nothing to test voriconazole resistance or susceptibility. The combination does not "overcome VOR resistance...", it is simply equally effective against voriconazole resistant isolates as it is against voriconazole susceptible isolates. Is there a reason to believe this would not have been the case? Are there links between triazole and caspofungin resistance/susceptibility? If so, this would be very informative to cover in the Discussion (which, as written, is more of a restating of the Results rather than a Discussion) and would place these results in some more impactful context.
4. For Fig 2D: The data again do not support the conclusions, as stated. These experiments do not actually measure "...reduction of biofilm formation..." as stated on Page 7. The results simply report on the metabolic activity of a preformed biofilm. The data provided only show results of the higher concentrations of PPTN at 40 and 80 uM. Did the authors measure inhibition of biofilm metabolic activity at PPTN concentrations of 20 uM? If not, they should comment on this and not that, as would be expected, biofilm-phase growth requires PPTN higher drug levels for inhibition than are needed for planktonic cells. This would be another interesting point to address in the Discussion.
5. For Fig 3C: The conclusions are, again, not supported well by the data presented. The authors state on Page 8/9 that "...the combination treatment leads to increased cell death in *A. fumigatus* through effects on membrane permeability." The simple propidium iodide assay completed here cannot differentiate the effects of cell death versus permeability (i.e., are they permeabilized but not yet dead? Or, just dead and therefore permeable?). A separate viability assay must be run under these same conditions, especially given that the previous viability assays reported herein are run under completely different conditions (i.e., conidia vs germlings, time of drug exposure, etc). This is very important as the major conclusions of the manuscript (even offered in the abstract) is that PPTN affects membrane permeability. There is little-to-no data offered in this regard provided at all to support conclusion that increased permeability. The authors even state in the abstract that PPTN "...alters organization of the *A. fumigatus* cell membrane...". What data is provided to support this statement?
6. For Fig 7: Again, conclusions are not fully supported by the data provided. The authors state on Page 12 that "When mammalian cytotoxicity was examined with the compound combination, no enhanced PPTN or PPTN-NC cytotoxicity was observed relative to PPTN or PPTN-NC alone". Fig 7B clearly does not support this at higher concentrations of PPTN-NC. The authors also state on page 12 that, "...the combination of PPTN with CAS has minimal toxicity to mammalian cells...". However, Fig 7C show that, at the PPTN concentration that no significant enhancement of conidial clearance is seen in the infected A549 cells (Fig 7E), half of the HepG2 cells are dead/metabolically inactive. Further, at the PPTN concentration where significant clearance enhancement is seen, the HepG2 cells are completely dead/inactive. This needs to be acknowledged by the authors. For Fig 7A and B, data should be reported as "metabolic activity" or "XTT fluorescence" and not as "cell viability".
7. For Fig 7E: Why do the authors now use such high CAS concentrations? These are not at all physiological and are far above the concentrations used in all other assays throughout the study.

Reviewer #3 (Comments for the Author):

This manuscript provides the characterization of a repurposed drug recovered as a potential antifungal adjuvant therapy. This compound 4 - [4 - (4 - Piperidiny) phenyl] - 7 - [4 - (-(trifluoromethyl) phenyl)] - 2 naphthalenecarboxylic acid (PPTN) has adjuvant activity on several antifungal agents and impacts a variety of different fungi, most prevalently here *Aspergillus fumigatus*. PPTN enhanced echinocandin and azole action on *A. fumigatus* voriconazole- and caspofungin-resistant isolates as well as to increase echinocandin activity against *Candida* spp and *Cryptococcus neoformans*. A haploinsufficiency screen conducted in the yeast *Saccharomyces cerevisiae* suggests genes involved in chromatin regulation could affect PPTN potency. Consistent with this suggestion, some chemical inhibitors of histone modifications also interacted with PPTN. *A. fumigatus* cells were analyzed by

RNA-seq in cells grown in caspofungin, PPTN and both compounds together. Greater than 200 genes were upregulated in the presence of both compounds while 400 were downregulated including ABC and other transporters. Toxicity of PPTN and caspofungin was modest in an epithelial cell model and slightly higher in HepG2 cells. The combination of these two compounds was more efficacious at clearing *A. fumigatus* cells from infected A549 (epithelial cells).

The authors carried out a large amount of work in this characterization of PPTN. Both of the labs involved are experts in this area and the work is well-done. I found the manuscript quite straightforward and clear.

My only minor suggestion is to provide whatever genotypic information is available for the *A. fumigatus* strains in Table 1 that are listed as voriconazole resistant. Do all these contain changes in the *cyp51A* gene or are other mechanisms of resistance potentially involved?

Reviewer #1:

This manuscript describes the characterization of a compound that was previously reported by the investigative group to decrease the MEC of caspofungin when used in combination against *A. fumigatus*. Here, they report on the activity of this compound, alone and in combination with CAS and other antifungals, against a larger selection of *Aspergillus* and other fungal isolates. The authors also provide cell biological and transcriptional studies to explore MOA of the compound and provide data regarding toxicity. In general, the studies are interesting and carry impact for the field. However, there are multiple issues with conclusions not being supported by the data and additional clarity of experimental approaches that are needed.

Answer: We thank the reviewer for the excellent suggestions and comments.

An overall issue that the authors need to consider is that caspofungin treatment may actually be potentiating the activity of PPTN, and not the other way around as they conclude throughout the manuscript and in the title. There is not much evidence provided that this is not, in fact, the case. PPTN does have some intrinsic antifungal activity as the authors report that it has measurable MICs against some *Aspergillus* isolates and against other pathogenic fungi. Also, PPTN showed synergy/additive effects with all of the antifungal used. The triazoles and polyenes caused weakened membranes. It would make more sense that each of these antifungals allow accumulation of PPTN in the fungal cell to enhance the antifungal effect of PPTN itself. Therefore, the antifungals potentiate PPTN. What evidence truly shows that PPTN potentiates the antifungals? It would've been extremely helpful for the authors to have included line numbers. I have tried to utilize the appropriate page numbers below to assist the authors in locating the pertinent areas of the manuscript.

Answer: We thank the reviewer for this excellent observation. The reviewer is absolutely right about the fact that we cannot be sure if PPTN is potentiating CAS or if CAS is potentiating PPTN. We have changed the title and along the manuscript this concept replacing it by the idea they are interacting or synergizing. We apologize about not adding the line numbers to the manuscript. They were now added.

1. For Fig 1b: This data should not be expressed as "% of viable conidia". Due to the hyphal growth that occurs and the may these hyphae break apart into an unpredictable number of viable CFUs during removal from the 96-well plates and subsequent plating for calculations, there is no way to conclude that these values are predicative of the viability of conidia. This should just be expressed as CFU relative to untreated controls.

Answer: We thank the reviewer for the observation. We have changed the legend of the y-axis to: "CFUs relative to untreated controls"

2. For Fig 1C: The authors need to clarify their wording related to how they describe this data. On Page 6, the authors state that "...the fungal cells were metabolically active up to 0.5 ug/ml (of CAS)". There are a wealth of publications

showing there is no 100% MIC for CAS against *Aspergillus* species and, therefore, the cells are certainly metabolically active at this drug concentration. The inability to detect "metabolic activity" here is just a limitation of the assay.

Answer: We thank the reviewer for the suggestion. The sentence was removed from the manuscript.

3. For Table 1: The data do not support the conclusions, as stated. The authors state on Page 7, that "These results suggest that the combination of PPTN with CAS can overcome both CAS and VOR resistance...". However, the authors do nothing to test voriconazole resistance or susceptibility. The combination does not "overcome VOR resistance...", it is simply equally effective against voriconazole resistant isolates as it is against voriconazole susceptible isolates. Is there a reason to believe this would not have been the case? Are there links between triazole and caspofungin resistance/susceptibility? If so, this would be very informative to cover in the Discussion (which, as written, is more of a restating of the Results rather than a Discussion) and would place these results in some more impactful context.

Answer: We thank the reviewer for the suggestion. The main idea is that PPTN+CAS can be used against VOR-resistant isolates as efficiently as against VOR-susceptible isolates. The sentence was changed to (Page 7, lines 210 to 212): "These results suggest that the combination of PPTN with CAS is equally effective against *A. fumigatus* voriconazole resistant isolates as it is against voriconazole susceptible isolates."

4. For Fig 2D: The data again do not support the conclusions, as stated. These experiments do not actually measure "...reduction of biofilm formation..." as stated on Page 7. The results simply report on the metabolic activity of a preformed biofilm. The data provided only show results of the higher concentrations of PPTN at 40 and 80 μ M. Did the authors measure inhibition of biofilm metabolic activity at PPTN concentrations of 20 μ M? If not, they should comment on this and not that, as would be expected, biofilm-phase growth requires PPTN higher drug levels for inhibition than are needed for planktonic cells. This would be another interesting point to address in the Discussion.

Answer: We thank the reviewer for this important observation. We have modified the sentences (Page 7, lines 216 to 223): "While the addition of the single agents alone resulted in no significant reduction of metabolic activity of the biofilm formation (**Figure 2d**), the combination of CAS with PPTN resulted in significant reductions of the metabolic activity of a preformed biofilm relative to individual treatments (**Figure 2d**). Interestingly, biofilm-phase growth requires PPTN higher drug levels for inhibition than are needed for planktonic cells. Thus, CAS and PPTN are effective at inhibiting *A. fumigatus* growth under both planktonic and biofilm conditions."

5. For Fig 3C: The conclusions are, again, not supported well by the data presented. The authors state on Page 8/9 that "...the combination treatment leads to increased cell death in *A. fumigatus* through effects on membrane permeability." The simple propidium iodide assay completed here cannot

differentiate the effects of cell death versus permeability (i.e., are they permeabilized but not yet dead? Or, just dead and therefore permeable?). A separate viability assay must be run under these same conditions, especially given that the previous viability assays reported herein are run under completely different conditions (i.e., conidia vs germlings, time of drug exposure, etc). This is very important as the major conclusions of the manuscript (even offered in the abstract) is that PPTN affects membrane permeability. There is little-to-no data offered in this regard provided at all to support conclusion that increased permeability. The authors even state in the abstract that PPTN "...alters organization of the *A. fumigatus* cell membrane...". What data is provided to support this statement?

Answer: We thank the reviewer for the comments. The reviewer is absolutely right. The observations related to a possible cell membrane permeability defect caused by PPTN+CAS were removed from the manuscript, including the abstract section.

6. For Fig 7: Again, conclusions are not fully supported by the data provided. The authors state on Page 12 that "When mammalian cytotoxicity was examined with the compound combination, no enhanced PPTN or PPTN-NC cytotoxicity was observed relative to PPTN or PPTN-NC alone". Fig 7B clearly does not support this at higher concentrations of PPTN-NC. The authors also state on page 12 that, "...the combination of PPTN with CAS has minimal toxicity to mammalian cells...". However, Fig 7C show that, at the PPTN concentration that no significant enhancement of conidial clearance is seen in the infected A549 cells (Fig 7E), half of the HepG2 cells are dead/metabolically inactive. Further, at the PPTN concentration where significant clearance enhancement is seen, the HepG2 cells are completely dead/inactive. This needs to be acknowledged by the authors.

Answer: We thank the reviewer for the observation. We modified the whole section aiming to address the reviewer's concerns (Pages 11 and 12, lines 358 to 360 and 361 to 367, respectively): "When mammalian cytotoxicity was examined with compound combination in A549 epithelial cells, no enhance toxicity was observed at PPTN or NC-PPTN 20 or 40 μ M+CAS 100 μ g/mL (**Figures 7a and 7b**). In contrast, in A549 cells, combination of PPTN or NC-PPTN 80, 160 or 320 μ M+CAS 100 μ g/mL have a metabolic activity decrease of 25 % and 80 %, respectively (**Figures 7a and 7b**). In HepG2 cells, PPTN 10, 20, 40, and 80 μ M+CAS 50 μ g/mL showed a metabolic decrease of 20, 50, 80, and 90 %, respectively (**Figure 7c**). NC-PPTN+CAS 50 μ g/mL was less toxic to the HepG2 cells with no decreased metabolic activity in 10 and 20 μ M, but with 75 % decreased metabolic activity in 40 μ M (**Figure 7d**)."

For Fig 7A and B, data should be reported as "metabolic activity" or "XTT fluorescence" and not as "cell viability".

Answer: We thank the reviewer for this observation. "Cell viability" for mammalian cells were replaced along the manuscript by "metabolic activity".

7. For Fig 7E: Why do the authors now use such high CAS concentrations? These are not at all physiological and are far above the concentrations used in all other assays throughout the study.

Answer: We used such high concentrations of CAS because lower concentrations did not provide decreased conidial viability.

Reviewer #3:

This manuscript provides the characterization of a repurposed drug recovered as a potential antifungal adjuvant therapy. This compound 4 - [4 - (4 - Piperidinyl) phenyl] - 7 - [4- -(trifluoromethyl) phenyl] - 2 naphthalenecarboxylic acid (PPTN) has adjuvant activity on several antifungal agents and impacts a variety of different fungi, most prevalently here *Aspergillus fumigatus*. PPTN enhanced echinocandin and azole action on *A. fumigatus* voriconazole- and caspofungin-resistant isolates as well as to increase echinocandin activity against *Candida* spp and *Cryptococcus neoformans*. A haploinsufficiency screen conducted in the yeast *Saccharomyces cerevisiae* suggests genes involved in chromatin regulation could affect PPTN potency. Consistent with this suggestion, some chemical inhibitors of histone modifications also interacted with PPTN. *A. fumigatus* cells were analyzed by RNA-seq in cells grown in caspofungin, PPTN and both compounds together. Greater than 200 genes were upregulated in the presence of both compounds while 400 were downregulated including ABC and other transporters. Toxicity of PPTN and caspofugin was modest in an epithelial cell model and slightly higher in HepG2 cells. The combination of these two compounds was more efficacious at clearing *A. fumigatus* cells from infected A549 (epithelial cells).

The authors carried out a large amount of work in this characterization of PPTN. Both of the labs involved are experts in this area and the work is well-done. I found the manuscript quite straightforward and clear.

Answer: We thank the reviewer for the comments.

My only minor suggestion is to provide whatever genotypic information is available for the *A. fumigatus* strains in Table 1 that are listed as voriconazole resistant. Do all these contain changes in the *cyp51A* gene or are other mechanisms of resistance potentially involved?

Answer: We thank the reviewer for the excellent suggestion. We have added this information to the Table 1 and to the text (Page 7, lines 195 to 198): “Subsequently, we assessed if the combination of CAS with PPTN would be effective against *A. fumigatus* CAS- or voriconazole (VOR)-resistant *CYP51*-dependent and –independent clinical isolates (that have VOR MICs >2.0 µg/mL; **Table 1**)”.

Re: Spectrum03318-24R1 (The GPCR antagonist PPTN synergizes with caspofungin providing increased fungicidal activity against *Aspergillus fumigatus*)

Dear Gustavo :

I am happy to say that your revised manuscript has been favorably reviewed by the reviewer. They thought your revisions were very helpful.

Your manuscript has been accepted, and I am forwarding it to the ASM production staff for publication. Your paper will first be checked to make sure all elements meet the technical requirements. ASM staff will contact you if anything needs to be revised before copyediting and production can begin. Otherwise, you will be notified when your proofs are ready to be viewed.

Sincerely,
James Konopka
Editor
Microbiology Spectrum

Reviewer #1 (Comments for the Author):

The authors have positively responded to all prior comments. The conclusions are now better aligned with the results, as provided.